# CORRELATION BETWEEN TECTONIC STRESS REGIMES AND METHANE SEEPAGE ON THE WEST-SVALBARD MARGIN

A. Plaza-Faverola[1] and M. Keiding[2]

[1] CAGE-Centre for Arctic Gas Hydrate, Environment, and Climate; Department of Geosciences, UiT The Arctic University of Norway, N-9037 Tromsø, Norway

[2] Geological Survey of Norway (NGU), P.O. Box 6315 Torgarden, 7491 Trondheim, Norway

*Correspondence to*: Andreia Plaza-Faverola (Andreia.a.faverola@uit.no)

**Abstract.** Methane seepage occurs across the west-Svalbard margin at water depths ranging from < 300 m, landward from the shelf break, to > 1000 m in regions just a few kilometres away from the mid-ocean ridges in the Fram Strait. The mechanisms controlling seepage remain elusive. The Vestnesa sedimentary ridge, located on oceanic crust at 1000-1700 m water depth, hosts a perennial gas hydrate and associated free gas system. The restricted occurrence of acoustic flares to the eastern segment of the sedimentary ridge, despite the presence of pockmarks along the entire ridge, indicates a spatial variation in seepage activity. This variation coincides with a change in the faulting pattern as well as in the characteristics of fluid flow features. Due to the position of the Vestnesa ridge with respect to the Molloy and Knipovich mid-ocean ridges, it has been suggested that seepage along the ridge has a tectonic control. We modelled the tectonic stress regime due to oblique spreading along the Molloy and Knipovich ridges to investigate whether spatial variations in the tectonic regime along the Vestnesa Ridge are plausible. The model predicts a zone of tensile stress that extends northward from the Knipovich Ridge and encompasses the zone of acoustic flares on the eastern Vestnesa Ridge. In this zone the orientation of the maximum principal stress is parallel to pre-existing faults. The model predicts a strike-slip stress regime in regions with pockmarks where acoustic flares have not been documented. If a certain degree of coupling is assumed between deep crustal and near-surface deformation, it is possible that ridge push forces have influenced seepage activity in the region by interacting with the pore-pressure regime at the base of the gas hydrate stability zone. More abundant seepage on the eastern Vestnesa Ridge at present may be facilitated by dilation of faults and fractures favourably oriented with respect to the stress field. A modified state of stress in the past, for instance due to more significant glacial stress, may have explained a vigorous seepage activity along the entire Vestnesa Ridge. The contribution of other mechanisms to the state of stress (i.e., sedimentary loading and lithospheric flexure) remain to be investigated. Our study provides a first order assessment of how tectonic stresses may be influencing the kinematics of near-surface faults and associated seepage activity offshore the west-Svalbard margin.

## 1. Introduction


Hundreds of gigatonnes of carbon are stored as gas hydrates and shallow gas reservoirs in continental margins
(e.g., Hunter et al., 2013). The release of these carbons over geological time, a phenomenon known as methane
seepage, is an important contribution to the global carbon cycle. Understanding and quantifying seepage has
important implications for ocean acidification, deep-sea ecology and global climate. Periods of massive methane
release from gas hydrate systems (e.g., Dickens, 2011) or from large volcanic basins like that in the mid-
Norwegian margin (e.g., Svensen et al., 2004) have been linked to global warming events such as the Palaeocene-
Eocene thermal maximum. In addition, methane seepage and near-seafloor gas migration have implications for
geohazards, since pore-fluid pressure destabilization is one factor associated with the triggering of submarine
land-slides (e.g., DeVore and Sawyer, 2016;Urlaub et al., 2015). It is well known that seepage at continental
margins has been occurring episodically for millions of years (e.g., Judd and Hovland, 2009), but there is a poor
understanding of what forces it.

Present day seepage is identified as acoustic flares in the water column commonly originating at seafloor
depressions (e.g., Chand et al., 2012;Salomatin and Yusupov, 2011;Skarke et al., 2014;Smith et al.,
2014;Westbrook et al., 2009), while authigenic carbonate mounds are used as indicators of longer-term seepage
activity (e.g., Judd and Hovland, 2009). Seepage at the theoretical upstream termination of the gas hydrate
stability zone (GHSZ) (i.e., coinciding with the shelf edge) at different continental margins, has been explained
by temperature driven gas-hydrate dissociation (e.g., Skarke et al., 2014;Westbrook et al., 2009). On formerly
glaciated regions off Svalbard and the Barents Sea, active seepage has been explained by gas hydrate dissociation
either due to pressure changes resulting from the retreat of the ice-sheet (e.g., Portnov et al., 2016;Andreassen et
al., 2017) or to post-glacial uplift (Wallmann et al., 2018).

Across the west-Svalbard margin, active seepage extends beyond the shelf break and the region formerly covered
by ice. As a matter of fact, active seepage sites have been identified from inside Isfjorden (Roy et al., 2014) to
water depths of ~1200 m (Smith et al., 2014) where the Vestnesa Ridge hosts a perennially stable gas hydrate
system > 50 km seaward from the ice-sheet grounding line. The Vestnesa Ridge is a NW-SE oriented contourite
deposit located between the northward termination of the Knipovich Ridge and the eastern flank of the Molloy
spreading ridge in the Fram Strait (Fig. 1). Seafloor pockmarks along the Vestnesa Ridge, first documented by
Vogt et al., (1994), exist along the entire ridge. However, acoustic flares have been observed to originate
exclusively at large pockmarks located on the eastern part of the sedimentary ridge (Fig. 2, 3). A clear increase in
seepage activity towards the easternmost part of the ridge is thus evident from multiple year's water-column
acoustic surveys (Petersen et al., 2010;Bünz et al., 2012;Plaza-Faverola et al., 2017;Smith et al., 2014). In this
paper, we use the terminology "active" and "inactive" to differentiate between sites with and without documented
acoustic flares. Even though methane advection and methanogenesis are likely to be active processes along the
entire Vestnesa Ridge, the presence of inactive pockmarks adjacent to a zone of active seepage, raises the
question what controls temporal and spatial variations in seepage activity along the ridge?

Plaza-Faverola et al., (2015) documented seismic differences in the orientation and type of faulting along the
ridge and showed a link between the distribution of gas chimneys and faults. They hypothesised that seepage
activity may be explained by spatial variation in tectonic stress field across the margin (Plaza-Faverola et al.,
2015). However, the state of stress across Arctic passive margins has not been investigated. The total state of
stress at formerly glaciated continental margins can be the result of diverse factors including bathymetry and
subsurface density contrasts, subsidence due to glacial or sedimentary loading and lithospheric cooling, in
addition to ridge-push forces (Fejerskov and Lindholm, 2000;Lindholm et al., 2000;Olesen et al., 2013;Stein et
al., 1989;Grunnaleite et al., 2009).

The interaction between the above mentioned factors renders modelling of the total state of stress a complex
problem that has not yet been tackled. In this study, we focus exclusively on the potential contribution of oblique
spreading at the Molloy and the Knipovich ridges to the total state of stress along the Vestnesa Ridge and do a
qualitative analysis of how stress generated by mid-ocean ridge spreading may influence near-surface faulting
and associated seepage activity. The study of the effect of ridge push forces on near-surface deformation across
the west-Svalbard margin contributes to the current debate about neo-tectonism and stress field variations across
passive margins (Olesen et al., 2013;Salomon et al., 2015). It also has implications for understanding the
mechanisms that control seepage at continental margins globally. Splay-faults are found to drive fluid migration
at subduction margins and to sustain shallow gas accumulations and seepage (e.g., Plaza-Faverola et al.,
2016;Minshull and White, 1989;Moore and Vrolijk, 1992;Crutchley et al., 2013), and the relationship between
fault kinematics and fluid migration has been documented specially at accretionary margins where earthquake-
induced seafloor seepage has been observed (e.g., Geersen et al., 2016). So far, the information about the present
day stress regime in the Fram Strait has been limited to large scale lithospheric density models (Schiffer et al.,
2018) and a number of stress vectors from earthquake focal mechanisms along the mid-Atlantic plate boundary
(Heidbach et al., 2016). Our study provides a first order assessment of how stresses from slow spreading mid-
ocean ridges may be influencing the kinematics of near-surface faults and associated seepage activity across an
Arctic passive margin.
## 2.  Structural and stratigraphic setting
In the Fram Strait, sedimentary basins are within tens of kilometres from ultra-slow spreading Arctic mid-ocean
ridges (Fig. 1). The opening of the Fram Strait was initiated 33 Ma ago and evolved as a result of slow spreading
of the Molloy and Knipovich Ridges (Engen et al., 2008). An important transpressional event deformed the
sedimentary sequences off western Svalbard, resulting in folds and thrustbelts, during the Paleocene-Eocene
dextral movement of Spitsbergen with respect to Greenland. Transpression stopped in the early Oligocene when
the tectonic regime became dominated by extension (Myhre and Eldholm, 1988). The circulation of deep water
masses through the Fram Strait started during the Miocene, ca. 17-10 Ma ago (Jakobsson et al., 2007;Ehlers and
Jokat, 2009), and established the environmental conditions for the evolution of bottom current-driven
sedimentary drifts (Eiken and Hinz, 1993;Johnson et al., 2015). It has been suggested that the opening of the
northern Norwegian–Greenland Sea was initiated by the northward propagation of the Knipovich ridge into the
ancient Spitsbergen Shear Zone (Crane et al., 1991) .
The continental crust beneath the western coast of Svalbard thins towards the Hornsund Fault zone indicating
extension following the opening of the Greenland Sea (Faleide et al., 1991). Late Miocene and Pliocene
sedimentation, driven by bottom currents, resulted in the formation of the ca. 100 km long Vestnesa Ridge
between the shelf break off west-Svalbard and oceanic crust highs at the eastern flank of the Molloy mid-ocean
ridge (Eiken and Hinz, 1993;Vogt et al., 1994). The sedimentary ridge is oriented parallel to the Molloy
Transform Fault and its crest experiences a change in morphology from narrow on the eastern segment to broader
on the western Vestnesa Ridge segment (Fig. 2). The exact location of the continental-ocean transition remains
uncertain (Eldholm et al., 1987) but it is inferred to be nearby the transition from the eastern to the western
segments (Engen et al., 2008).
The total sedimentary thickness along the Vestnesa Ridge remains unconstrained. Based on one available
regional seismic profile it can be inferred that the ridge is > 5 km thick in places (Eiken and Hinz, 1993). It has
been divided into three main stratigraphic units (Eiken and Hinz, 1993;Hustoft, 2009): the deepest sequence,
YP1, consists of synrift and post-rift sediments deposited directly on oceanic crust; YP2 consists of contourites;
and YP3, corresponding to the onset of Pleistocene glaciations (ca. 2.7 Ma ago) (Mattingsdal et al., 2014), is
dominated by glaciomarine contourites and a mix with turbidites in regions close to the shelf break. The effect of
ice-sheet dynamics on the west-Svalbard margin (Patton et al., 2016;Knies et al., 2009) has influenced the
stratigraphy, and most likely the morphology, of the Vestnesa Ridge and adjacent sedimentary basins. In this
Arctic region, glaciations are believed to have started even earlier than 5 Ma ago. The local intensification of
glaciations is inferred to have started ca. 2.7 Ma ago (e.g., Faleide et al., 1996;Mattingsdal et al., 2014). Strong
climatic fluctuations characterized by intercalating colder, intense glaciations with warmer and longer
interglacials, dominated the last ca. 1 Ma. (e.g., Jansen et al., 1990;Jansen and Sjøholm, 1991).

### 3. Seismic data
The description of faults and fluid flow related features along the Vestnesa Ridge is documented by several
authors (Bünz et al., 2012;Hustoft, 2009;Petersen et al., 2010;Plaza-Faverola et al., 2015;Plaza-Faverola et al.,
2017). Two-3D high resolution seismic data sets acquired on the western and the eastern Vestnesa Ridge
respectively (Fig. 2), and one 2D seismic line acquired along the entire Vestnesa Ridge extent have been
particularly useful in the description of the structures along the ridge (Fig. 2). These data have been previously
used for the investigation of the bottom simulating reflection dynamics (i.e., the seismic indicator of the base of
the gas hydrate stability zone) (Plaza-Faverola et al., 2017) and documentation of gas chimneys and faults in the
region (Petersen et al., 2010;Plaza-Faverola et al., 2015;Bünz et al., 2012). The 3D seismic data were acquired on
board R/V Helmer Hanssen using the high resolution P-Cable system (Planke et al., 2009). The 2D lines were
also collected connecting 4 streamers from the P-Cable system for 2D acquisition. Final lateral resolution of the
3D data sets is given by a bin size of 6.25x6.25 m$^2$ and the vertical resolution is > 3 m with a dominant frequency
of 130 Hz. Details about acquisition and processing can be found in Petersen et al., 2010 and Plaza-Faverola et
al., 2015. For the 2D survey the dominant frequency was ~80 Hz resulting in a vertical resolution > 4.5 m
(assumed as λ/4 with an acoustic velocity in water of 1469 m/s given by CTD data; Plaza-Faverola et al., 2017).

### 4. The modelling approach
The modelling carried out in this study deals exclusively with tectonic stress due to ridge push. We use the
approach by Keiding et al. (2009) based on the analytical solutions derived by Okada (1985), to model the plate
motion and tectonic stress field due to spreading along the Molloy and Knipovich Ridges.

The Okada model and our derivation of the stress field from it is described in more detail in appendix A. The
Molloy and Knipovich Ridges are modelled as rectangular planes with opening and transform motion in a flat
Earth model with elastic, homogeneous, isotropic rheology (Fig. A1 in appendix). Each rectangular plane is
defined by ten model parameters used to approximate the location, geometry and deformation of the spreading
ridges (Okada, 1985; see supplement Table 1). The locations of the two spreading ridges were constrained from
bathymetry maps (Fig. 1). The two spreading ridges are assumed to have continuous, symmetric deformation
below the brittle-ductile transition, with a half spreading rate of 7 mm/yr and a spreading direction of N125°E,
according to recent plate motion models (DeMets et al., 2010). Because the spreading direction is not
perpendicular to the trends of the spreading ridges, this results in both opening and right-lateral motion; that is,
oblique spreading on the Molloy and Knipovich Ridges. The Molloy Transform Fault, which connects the two
spreading ridges, trends N133°E, thus a spreading direction of N125°E implies extension across the transform
zone. We use a depth of 10 km for the brittle-ductile transition and 900 km for the lower boundary of the
deforming planes, to avoid boundary effects. For the elastic rheology, we assume typical crustal values of
Poisson's ratio = 0.25 and shear modulus = 30 GPa (Turcotte and Schubert, 2002). We perform sensitivity tests
for realistic variations in 1) model geometry, 2) spreading direction, 3) depth of the brittle-ductile transition, and
4) Poisson's ratio (Supplementary material). Variations in shear modulus, e.g. reflecting differences in elastic
parameters of crust and sediments, would not influence the results, because we do not consider the magnitude of
the stresses.

Asymmetric spreading has been postulated for the Knipovich Ridge based on heat flow data (Crane et al., 1991),
and for other ultraslow spreading ridges based on magnetic data (e.g., Gaina et al., 2015). However, the evidence
for asymmetry along the Knipovich Ridge remains inconclusive and debatable in terms, for example, of the
relative speeds suggested for the North American (faster) and the Eurasian (slower) plates (Crane et al.,
1991;Morgan, 1981;Vogt et al., 1994). This reflects that the currently available magnetic data from the west-
Svalbard margin is not of a quality that allows an assessment of possible asymmetry of the spreading in the Fram
Strait (Nasuti and Olesen, 2014). Symmetry is thus conveniently assumed for the purpose of the present study.

We focus on the stress field in the upper part of the crust (where the GHSZ is) and characterise the stress regime
based on the relationship between the horizontal and vertical stresses. We refer to the stresses as $\sigma_v$ (vertical
stress), $\sigma_H$ (maximum horizontal stress) and $\sigma_h$ (minimum horizontal stress), where compressive stress is positive
(Zoback and Zoback, 2002). A tensile stress regime ($\sigma_v > \sigma_H > \sigma_h$) favours the opening of steep faults that can
provide pathways for fluids. Favourable orientation of stresses with respect to existing faults and/or pore fluid
pressures increasing beyond hydrostatic pressures are additional conditions for leading to opening for fluids
under strike-slip ($\sigma_H > \sigma_v > \sigma_h$) and compressive ($\sigma_H > \sigma_h > \sigma_v$) regimes (e.g., Grauls and Baleix, 1994).

**5. Results**
**5.1 Predicted type and orientation of stress fields due to oblique spreading at the Molloy and the Knipovich**
**ridges**
The model predicts zones of tensile stress near the spreading ridges, and strike-slip at larger distances from the
ridges. An unexpected pattern of tensile stress arises from the northward termination and the southward
termination of the Knipovich and Molloy ridges respectively (Fig. 3). The zone of tensile stress that extends
northward from the Knipovich Ridge, encompasses the eastern part of the Vestnesa Ridge. The western Vestnesa
Ridge, on the other hand, lies entirely in a zone of strike-slip stress (Fig. 3). The sensitivity tests show that the
tensile stress zone is a robust feature of the model, that is, variations in the parameters result in a change of the
extent and shape of the tensile zone but the zone remains in place (Supplementary material). It appears that the
tensile zone is a result of the interference of the stress from the two spreading ridges. To illustrate this, we ran the
model for the Molloy Ridge and the Knipovich Ridge independently. In the model with Knipovich Ridge alone, a
large tensile zone extends northeast from the ridge's northern end, covering only the easternmost corner of
Vestnesa Ridge (Fig. 4). Under the influence of the strike-slip field from the Molloy Ridge, this zone is deflected
and split into two lobes, of which one extends to the eastern Vestnesa Ridge segment.

To investigate the geometric relationship between the predicted stress field and mapped faults, we calculated the
orientations of maximum compressive horizontal stress (Lund and Townend, 2007). The maximum horizontal
stresses ($\sigma_H$) approximately align with the spreading axes within the tensile regime and are perpendicular to the
axes within the strike-slip regime (Fig. 3). Spreading along the Molloy ridge causes NW-SE orientation of the
maximum compressive stress along most of the Vestnesa Ridge, except for the eastern segment where the
influence of the Knipovich Ridge results in a rotation of the stress towards E-W (Fig. 3).

The simplifying assumptions involved in our model imply that the calculated stresses in the upper crust are
unconstrained to a certain degree. However, the predicted stress directions are in general agreement with other
models of plate tectonic forces (e.g., Gölke and Coblentz, 1996;Naliboff et al., 2012). In addition, Árnadóttir et
al. (2009) demonstrated that the deformation field from the complex plate boundary in Iceland could be modelled

using Okada's models. More importantly, a comparison of the predicted strike-slip and tensile stress fields from plate spreading and observed earthquake focal mechanisms shows an excellent agreement, both with regards to stress regime and orientation of maximum compressive stress. The earthquake focal mechanisms are mostly normal along the spreading ridges and strike-slip along the transform faults, and the focal mechanism pressure axes align nicely with the predicted directions of maximum compressive stress (Fig. 3). The good agreement between Okada's model and other modelling approaches as well as between the resulting stresses and focal mechanisms in the area indicates that the model, despite the simplicity of its assumptions, provides a correct first order prediction of orientation and type of the stress field in the upper crust (other possible sources of stress in the region will be discussed in more detail in section 6.1). It remains an open question to which degree the crustal stresses are transferred to the sedimentary successions of the Vestnesa Ridge. For compacted stratigraphic formations in the Norwegian Sea, a comparison of shallow in-situ stress measurements and deeper observations from earthquake focal mechanisms indicates that the stress field is homogeneous in direction over a large depth range (Fejerskov and Lindholm, 2000). For an overburden constituted of Quaternary sediments, though, the stress coupling between the crust and the near-surface depends on the shear strength of the sediments. The upper 200 m of hemipelagic sediment along the Vestnesa Ridge are relatively young (< 2 Ma) and the degree of sediment consolidation remains uninvestigated. However, the fact that a large number of faults extend several hundred meters through the sediments suggests that compaction of the sediments has been large enough to build up some amount of shear strength. Geotechnical studies from different continental margins indicate that deep marine sediments can experience high compressibility due to the homogeneity in the grain structure (i.e., large areas made of a single type of sediment), providing favourable conditions for shear failure (Urlaub et al., 2015;DeVore and Sawyer, 2016). Therefore, we consider possible that the upper sedimentary column along the Vestnesa Ridge has been deformed by tectonic stress.

**5.2 Distribution of faults and seepage activity along the Vestensa Ridge with respect to modelled tectonic stress**

High-resolution 3D seismic data collected on the eastern Vestnesa Ridge revealed sub-seabed NW-SE oriented, near-vertical faults with a small normal throw (< 10 m; Fig. 5). In this part of the Vestnesa Ridge, gas chimneys and seafloor pockmarks are ca. 500 m in diameter. On structural maps extracted along surfaces within the GHSZ gas chimneys project over fault planes or at the intersection between fractures (Fig. 2, 3c). A set of N-S to NNE-SSE trending faults outcrop at the seafloor at a narrow zone between the Vestnesa Ridge and the northern termination of the Knipovich Ridge (Fig 1, 2). These faults have been suggested to indicate ongoing northward

propagation of the Knipovich rift system (Crane et al., 2001;Vanneste et al., 2005). The NW-SE oriented sub-
seabed faults and fractures at the crest of the Vestensa Ridge could be genetically associated with these
outcropping faults (Plaza-Faverola et al., 2015; Fig. 2).

Most of the outcropping N-S to NNE-SSE oriented faults north of the Knipovich Ridge and the sub-seafloor NW-
SE oriented faults on the eastern Vestnesa Ridge are located within the zone of modelled tensile regime that
extends northward from the Knipovich Ridge (Fig. 3). The orientation of $\sigma_H$ rotates from being perpendicular to
the Molloy ridge nearby sub-seafloor faults at the eastern Vestensa Ridge, to be more perpendicular to the
Knipovich Ridge in places within the tensile zone (Fig. 3). Interestingly, documented acoustic flares along the
Vestensa Ridge are also located within the zone of modelled tensile stress regime (Fig. 3). The match between the
extent of the modelled tensile regime and the active region of pockmarks is not exact; pockmarks with acoustic
flares exist a few kilometres westward from the termination of the tensile zone (Fig. 3). However, the agreement
is striking from a regional point of view. Some of the outcropping faults north of the Knipovich Ridge and south
of the Molloy transform fault appear located outside the modelled tensile zone (Fig. 3; Fig. S1-S4 in the
supplement). Inactive pockmarks (i.e., no acoustic flares have been observed during several visits to the area) are
visible on high resolution bathymetry maps over these faulted regions (Dumke et al., 2016;Hustoft, 2009;Johnson
et al., 2015;Waghorn et al., 2018).

In a similar high-resolution 3D seismic data set from the western Vestnesa Ridge the faults have different
characteristics compared to those of the eastern segment. In this part of the ridge gas chimneys are narrower,
buried pockmarks are stacked more vertically than the chimneys towards the east and it is possible to recognise
more faults reaching the present-day seafloor (Plaza-Faverola et al., 2015). Fault segments are more randomly
oriented with a tendency for WNW-ESE and E-W orientations (Fig. 2). These structures coincide with a
modelled strike-slip stress regime with $\sigma_H$ oriented nearly perpendicular to the Molloy Ridge (Fig. 3).

**6. Discussion**
The striking coincidence between the spatial variation in modelled stress regimes and the pattern of faulting and
seepage activity along the Vestnesa Ridge leads to the discussion whether tectonic stresses resulting from plate
spreading at the Molloy and the Knipovich ridges have the potential to influence near-surface deformation and
fluid dynamics in the study area. We discuss first the modelling results in the context of the total state of stress
across passive margins and to which extent regional stresses can influence near-surface deformation. Assuming
that tectonic stress can potentially influence near-surface deformation, we discuss then the effect that the
modelled stress fields would have on pre-existing faults and associated fluid migration. Finally, we propose a
model for explaining seepage evolution along the Vestnesa Ridge coupled to stress field variations. We close the
discussion with a note on the implications of the present study for understanding near-surface fluid dynamics
across passive margins globally.

**6.1 Modelled stress in the context of the state of stress along the Vestnesa Ridge**
In this study we focused exclusively on modelling the type and orientation of stresses potentially generated by
spreading at the Molloy and Knipovich ridges. Other sources of stress have been so far disregarded. Hence, the
modelled stress field documented in this study cannot be considered as a representation of the total stress field in
the region. Modelling studies from Atlantic-type passive margins, suggest that from all the possible sources of
stress across passive margins (i.e., sediment loading, glacial flexure, spatial density contrasts, and ridge push as
well as basal drag forces) sediment loading (assuming elastic deformation) appears to be the mechanism with the
potential of generating the largest magnitudes of stresses across passive margins (Stein et al., 1989;Turcotte et al.,
1977). However, stress information derived from seismological and in-situ data (Fjeldskaar and Amantov,
2018;Grunnaleite et al., 2009;Lindholm et al., 2000;Olesen et al., 2013) and paleo-stress field analyses based on
dip and azimuth of fault planes (Salomon et al., 2015) point towards a dominant effect of ridge push forces on the
state of stress across passive continental margins. Given the proximity of the Vestnesa Ridge to the Molloy and
the Knipovich ridges (Fig. 1), we argue that tectonic stress from spreading can be an important factor, perhaps
even a dominant factor, controlling near-surface deformation along the Vestnesa Ridge.

The contemporary stress field across the west-Svalbard passive margin is presumably the result of an interaction
between large-scale tectonic stress mechanisms (i.e., ridge push, basal drag) overprinted by regional (i.e., density
contrasts, glacial related flexure, sediment loading) and local mechanisms (e.g., topography, pore-fluid pressure
variations, faulting). In the concrete case of the Vestnesa Ridge, a change in the faulting pattern, the distribution
of shallow gas and gas hydrates, as well as differences in the topographic characteristics of the ridge crest (Fig. 2,
5), are all factors likely to induce local changes in the degree of compaction and in near-surface stress. We
discuss in the following sections how local stress-generating mechanisms may interact with tectonic forcing to
control fluid dynamics and seepage.

The Vestnesa sedimentary Ridge sits over relatively young oceanic crust, < 19 Ma old (Eiken and Hinz,
1993;Hustoft, 2009). The oceanic-continental transition is not well constrained but its inferred location crosses
the Vestnesa Ridge at its easternmost end (Engen et al., 2008;Hustoft, 2009). This is a zone prone to flexural
subsidence due to cooling during the evolution of the margin and the oceanic crust may have experienced syn-
sedimentary subsidence focused around the oceanic-continental transition, as suggested for Atlantic passive
margins (Turcotte et al., 1977). However, syn-sedimentary subsidence would result in N-S oriented faults (i.e.,
reflecting the main direction of major rift systems during basin evolution) (Faleide et al., 1991;Faleide et al.,
1996). Although one N-S oriented fault outcrops in bathymetry data at the transition from the eastern to the
western Vestnesa Ridge segments (Fig. 5a), most of the sub-seabed faults and associated fluid migration features
in 3D seismic data are NW-SE to E-W oriented (Fig. 1, 2).
The weight of the contourite ridge over the oceanic crust may have generated additional stress on the Vestnesa
Ridge. Sedimentation rates on the Vestnesa Ridge have been moderate, estimated to have fluctuated between 0.1-
0.6 mm/year since the onset of glaciations 2.7 Ma ago (Plaza-Faverola et al., 2017;Knies et al., 2018;Mattingsdal
et al., 2014). The lithology of the upper sediment along the ridge appears dominated by soft fine-grained
hemipelagic clayey silt with variable concentrations of ice-rafted debris (Sztybor and Rasmussen, 2017a).
Together, sedimentation rates and a high clay content would provide an ideal setting for undercompaction due to
increased pore fluid pressure (e.g., Fertl, 1976;Smith, 1999). High pore fluid pressure would lead to a decrease in
the effective stress and favour shearing (Grauls and Baleix, 1994). Whether these sedimentation rates have
allowed stress to build up through the upper strata faster than what it relaxes at the crust (i.e., as expected for
sedimentation rates larger than 1 mm/year (Stein et al., 1989)), as well as what has been the role of gas hydrates
and authigenic carbonate on the compaction history of the sediment remains to be investigated.
Glacial isostasy results in significant stresses associated with flexure of the lithosphere as the ice-sheet advances
or retreats. Present uplift rates are highest at the centre of the formerly glaciated region where the ice thickness
was at the maximum (Fjeldskaar and Amantov, 2018). Modelled present day uplift rates at the periphery of the
Barents sea ice-sheet ranges from 0 to -1 mm/year, depending on the ice-sheet model used in the calculation
(Auriac et al., 2016). This compares to an uplift rate of up to 9 mm/year at the centre of the ice sheet (Auriac et
al., 2016;Patton et al., 2016). Modelled glacial stresses induced by the Fennoscandian ice sheet on the mid-
Norwegian margin are close to zero at present day (Lund et al., 2009;Steffen et al., 2006). By analogy, present
day stress along the Vestnesa Ridge - located ~60 km from the shelf break - may be insignificant. It is likely that
glacial stresses as far off as the Vestnesa Ridge had a more significant effect in the past, as further discussed in
section 6.3 and 6.4.

Finally, ridge push forcing has the potential of being a dominant factor on the state of stress across the west-
Svalbard margin as observed for Norwegian margins (Fejerskov and Lindholm, 2000;Lindholm et al., 2000).
Specifically, the Vestnesa Ridge has the particularity that it is located within the expected range of maximum
influence of ridge push forces on the stress regime (Fejerskov and Lindholm, 2000) and that forces from two
spreading ridges influence it from different directions (i.e., the Molloy Ridge from the west and the Knipovich
Ridge from the south-east). The intriguing stress pattern appears to be caused by the interaction of the stress
generated by the two spreading ridges, as described above (section 5.1).

**6.2 Effect of the modelled stress fields on pre-existing faults and present day seepage**
Bearing in mind that several factors contribute to the total state of stress at different scales across passive margins
we assume that an influence on near-surface deformation by mid-ocean ridge stresses is plausible and discuss
their potential effect on seepage activity. Depending on the tectonic regime, the permeability through faults and
fractures may be enhanced or inhibited (e.g., Sibson, 1994;Hillis, 2001;Faulkner et al., 2010). Thus, spatial and
temporal variations in the tectonic stress regime may control the transient release of gas from the seafloor over
geological time as documented, for example, for $CO_2$ analogues in the Colorado Plateau (e.g., Jung et al., 2014).

A gas hydrate system is well developed and shallow gas accumulates at the base of the GHSZ along the entire
Vestnesa Ridge (Plaza-Faverola et al., 2017). Thermogenic gas accumulations at the base of the GHSZ (Fig. 5)
are structurally controlled (i.e., the gas migrates towards the crest of the sedimentary ridge) and together with
microbial methane this gas sustains present day seepage activity (Bünz et al., 2012;Plaza-Faverola et al.,
2017;Knies et al., 2018). However, seepage is focused and restricted. Some of the mechanisms commonly
invoked to explain seepage activity across passive margins include climate related gas hydrate dissociation, tidal
or seasonal sea-level changes, and pressure increases in shallow reservoirs due to fast sedimentation (e.g., Bünz
et al., 2003;Hustoft et al., 2010;Karstens et al., 2018;Riboulot et al., 2014;Skarke et al., 2014;Berndt et al.,
2014;Wallmann et al., 2018;Westbrook et al., 2009;Franek et al., 2017). While all of these mechanisms may
influence seepage systems as deep as the Vestnesa Ridge (> 1000 m deep; as discussed further in section 6.3)
they offer no explanation as to why seepage activity is more substantial within chimney sites proximal to or at
fault planes and why seepage is at a minimum or stopped elsewhere along the ridge (Fig. 2, 5). Overall, the

pattern of seepage activity along the Vestnesa Ridge is strikingly consistent with the modelled tectonic stress field pattern. Acoustic flares have been documented to originate from < 10 m broad zones (Panieri et al., 2017) within pockmarks located exclusively along faults. We suggest that these faults are favourably oriented with respect to a tectonic $\sigma_H$ (Fig. 2) and that opening of fault segments favourably oriented with respect to the stress field is one controlling factor of present day seepage.

Present day seepage activity is less pronounced towards the western Vestnesa Ridge. Despite available gas trapped at the base of the GHSZ (Fig. 5) the faults are generally less favourably oriented for tensile opening (i.e., NW-SE oriented $\sigma_H$) and are under a strike-slip regime (Fig. 2). The cluster of larger scale N-S to NNW-SSE trending extensional faults that outcrop at the southern slope of the Vestnesa Ridge (Fig. 1, 2), also coincides with the zone of predicted tensile stress (Fig. 3). However, the modelled maximum compressive stress in this area is generally oblique to the fault planes, making these faults less open for gas. Interestingly, this is also a zone of pockmarks where acoustic flares have not been observed (e.g., Johnson et al., 2015; Hustoft et al., 2009; Vanneste et al., 2005). A set of N-S oriented structures south of the Molloy Transform Fault and a train of pockmarks at the crest of a ridge west of the Knipovich Ridge axis are located under a strike-slip regime with N-S oriented $\sigma_H$ (Fig. 3). Although gas accumulations and gas hydrates have been identified at the crest of this ridge, acoustic flares have so far not been documented (Johnson et al., 2015; Waghorn et al., 2018). We suggest that the N-S trending faults in this region may be impermeable for fluids despite a parallel $\sigma_H$, if the stress regime is transpressive. Transpression has been documented at different stages of opening of the Fram Strait (Jokat et al., 2016; Myhre and Eldholm, 1988) and is thus a plausible tectonic mechanism for holding the gas from escaping. Ongoing studies will shed light into the structural evolution of this near-surface system.

The bathymetry of the southern flank of the Vestnesa Ridge deepens from 1200-1600 m along the crest of the Vestnesa Ridge to ca. 2000 m near the Molloy Transform Fault (Fig. 1). Thus, an additional effect of gravitational stress on near-surface deformation and seepage in the region cannot be ruled out. In particular, although the faults at the steep slope north of the Knipovich Ridge have been suggested to reflect the northward propagation of the Knipovich Ridge rift system (Crane et al., 2001; Vanneste et al., 2005), it is likely that their formation was influenced by gravitational stresses. Small-scale slumps at the slope (Fig 1, 2) could be also evidence of gravitational forcing at the steep southern flank of the Vestnesa Ridge. However, sub-seabed faults on the eastern Vestnesa Ridge dip towards the NE (Fig. 5c), suggesting that gravitational forcing is not necessarily influencing the behaviour of faults and current seepage activity on the eastern Vestensa Ridge.

403

### 6.3 Seepage evolution coupled to stress field variations

The seepage systems along the Vestnesa Ridge has been highly dynamic over geological time. Both microbial and thermogenic gas contribute to the gas hydrate and seepage system (Hong et al., 2016;Panieri et al., 2017;Plaza-Faverola et al., 2017;Smith et al., 2014). Reservoir modelling shows that source rock deposited north of the Molloy Transform Fault has potentially started to generate thermogenic gas 6 Ma ago and that migrating fluids reached the Vestnesa Ridge crest at the active seepage site ca. 2 Ma ago (Knies et al., 2018). Seepage has been occurring, episodically, at least since the onset of the Pleistocene glaciations directly through faults, and a deformation typical of gas chimneys (i.e., where periodicity is evidenced by buried pockmarks and authigenic carbonate crusts) seems to have started later (Plaza-Faverola et al., 2015). However, the periodicity of seepage events documented since the Last Glacial Maximum seems to correlate indistinctively with glacials or interglacials (Consolaro et al., 2015;Schneider et al., 2018a;Sztybor and Rasmussen, 2017b). One transient event was dated to ca. 17.000 years based on the presence of a ~1000 years old methane-dependent bivalve community possibly sustained by a gas pulse through a fault or chimney (Ambrose et al., 2015). A tectonic control on the evolution of near-surface fluid flow systems and seepage along the Vestensa Ridge is an explanation that reconciles the numerous cross-disciplinary observations in the area.

The spatial relation between gas chimneys at the crest of the ridge and fault planes (Fig. 2, 5c) (Bünz et al., 2012;Plaza-Faverola et al., 2015) is intriguing and raises the question whether the faulting was posterior to brecciation (fracturing) of the strata during chimney formation. Gas chimneys form by hydrofracturing generated at a zone of overpressure in a reservoir (e.g., Karstens and Berndt, 2015;Hustoft et al., 2010 and references therein;Davies et al., 2012). From the mechanical point of view the tensile faults at the eastern Vestnesa Ridge would not be a favourable setting for the generation of hydrofracturing and chimney formation right through fault planes as observed in the seismic (Fig. 2, 5c). For gas chimneys to be the youngest features fault segments would have to become tight and permeable at certain periods of times, allowing pore fluid pressure e.g., at the free gas zone beneath the GHSZ to build up (Fig. 5). This is a plausible scenario. The faults may get locally plugged with gas hydrates and authigenic carbonate and activate a self-sealing mechanism similar to that suggested for chimneys at other margins (e.g., Hovland, 2002). A model of gas hydrate-sealed faults and increased free gas zone underneath, has been suggested to explain seismic attenuation and velocities from an ocean bottom seismic experiment over the gas hydrate system north of the Knipovich Ridge (Madrussani et al., 2010). Nevertheless, where gas chimneys do not disturb the seismic response, fault planes are observed to extend near the seafloor

(Fig. 5c). This observation suggests that latest faulting periods may have broken through already brecciated
regions connecting gas chimneys that were already in place. Both cases are consistent with the fact that acoustic
flares and seepage bubbles are restricted to focused weakness zones (Panieri et al., 2017). We suggest that an
interaction between pore fluid pressure at the base of the GHSZ and tectonic stress has led to local stress field
variations and controlled seepage evolution. Opening of fractures is facilitated if the minimum horizontal stress is
smaller than the pore-fluid pressure ($p_f$), that is, the minimum effective stress is negative ($\sigma_h' = \sigma_h - p_f < 0$) (e.g.,
Grauls and Baleix, 1994). Secondary permeability may increase by formation of tension fractures near damaged
fault zones (Faulkner et al., 2010). Cycles of negative minimum effective stress and subsequent increase in
secondary permeability in a tensile stress regime can be achieved particularly easy in the near-surface and would
provide an explanation for the development of chimneys coupled to near-surface tectonic deformation. A
constant input of thermogenic gas from an Eocene reservoir since at least ca. 2 Ma ago would have contributed to
localized pore-fluid pressure increases (Knies et al., 2018).

Geophysical and paleontological data indicate that there was once more prominent seepage and active chimney
development on the western Vestnesa Ridge segment (e.g., Consolaro et al., 2015;Plaza-Faverola et al.,
2015;Schneider et al., 2018b). An interaction between pore-fluid pressure and tectonic stress would explain
variations in the amount of seepage activity over geological time. Following the same explanation as for the
present day seepage, the negative $\sigma_h'$ condition could have been attained anywhere along the Vestensa Ridge in
the past due to pore fluid pressure increases at the base of the GHSZ or due to favourable stress conditions.
During glacial periods, flexural stresses should have been significantly higher than at present day (Lund and
Schmidt, 2011). According to recent models of glacial isostasy by the Barents Sea Ice sheet during the last glacial
maximum, the Vesntesa Ridge laid in a zone where subsidence could have been of tens of meters (Patton et al.,
2016). At other times, before and after glacial maximums, the Vestnesa Ridge was possibly located within the
isostatic forebulge.

In general, it is expected that glacial-induced maximum horizontal stresses ($\sigma_H$) would be dominantly oriented
parallel to the shelf break (Björn Lund personal communication; Lund et al., 2009), that is, oriented N-S in the
area of the Vestnesa Ridge (Fig. 1). Such stress orientation would not favour opening for fluids along pre-exiting
NW-SE oriented faults associated with seepage activity at present (i.e., N-S oriented faults would be the more
vulnerable for opening). It is possible, though, that the repeated waxing and waning of the ice sheet caused a
cyclic modulation of the stress field (varying magnitude and orientation) and influenced the dynamics of gas
accumulations and favourably oriented faults along the Vestnesa Ridge in the past. Past glacial stresses may
provide then an alternative explanation for seepage along the entire Vestensa Ridge extent at given periods of
time (Fig. 6). This explanation is in line with the correlation between seepage and glacial-interglacial events
postulated for different continental margins e.g., for chimneys off the mid-Norwegian margin (Plaza-Faverola et
al., 2011), the Gulf of Lion (Riboulot et al., 2014), but also along the Vestnesa Ridge (Plaza-Faverola et al.,
2015;Schneider et al., 2018b).

A temporal variation in the stress field along the Vestnesa Ridge is also caused by its location on a constantly
growing plate. As the oceanic plate grows, the Vestnesa Ridge moves eastward with respect to the Molloy and
Knipovich Ridges, causing a westward shift in the regional stress field on the Vestnesa Ridge (Fig. 7). In future,
the eastern Vestnesa Ridge may temporarily move out of the tensile zone, while the western Vestnesa Ridge
moves into it (Fig. 7). This suggests that a negative effective stress and subsequent active seepage may reappear
and "reactivate" pockmarks to the west of the currently active seepage zone.

**6.4 Implications for the understanding of near-surface deformation across passive margins**
Our study is a first step in the investigation of the effect of regional stress on the dynamics of near-surface fluid
flow systems across passive margins. Analytical modelling of spreading at the Molloy and the Knipovich ridges
shows that complex stress fields may arise from the interaction of the dynamics at plate boundaries and exert an
effect across passive margins. Although the Vestnesa Ridge is a unique case study due to its remarkable
proximity to the Arctic mid-ocean ridges, stresses generated by plate tectonic forces are expected to extend for
thousands of km (Fejerskov and Lindholm, 2000). Across a single passive margin a range of regional and local
factors may result in spatial stress field variations that can explain focusing of gas seepage at specific regions. For
instance, the pervasive seepage zone west of Prins Karls Forland (PKF) on the west-Svalbard margin (Fig. 1)
could be under a stress regime that has been influenced by glacial rebound at a larger degree than at the Vestnesa
Ridge area over geological time. Wallmann et al., (2018) suggested that post glacial uplift lead to gas hydrate
dissociation after the Last Glacial Maximum and that such gas continues to sustain seepage off PKF. Previously,
several other studies argued for a gas-hydrate control on seepage in this region (e.g., Berndt et al., 2014;Portnov
et al., 2016;Westbrook et al., 2009). Since no gas hydrates have been found despite deep drilling (Riedel et al.,
2018) the gas hydrate hypotheses remain debatable. The influence of regional stresses on sub-seabed faults
suspected to underlay the seepage system (e.g., Mau et al., 2017) and shallow gas reservoirs (Knies et al., 2018)
provides an alternative and previously not contemplated explanation for seepage in this area. The interactions
between tectonic stress regimes and pore-fluid pressure we propose for explaining seepage evolution along the
Vestnesa Ridge may be applicable to seepage systems along other passive margins, in particular along Atlantic
passive margins where leakage from hydrocarbon reservoirs is prominent (e.g., the mid-Norwegian margin, the
Barents Sea, the North Sea, the north-east Greenland margin, the Mediterranean and even the Scotia plate
between Argentina and Antarctica) (e.g., Andreassen et al., 2017;Bünz et al., 2003;Hovland and Sommerville,
1985;Riboulot et al., 2014;Somoza et al., 2014;Vis, 2017). The Vestnesa Ridge case study adds a new perspective
to the current debate about the inactivity of passive margins (Fejerskov and Lindholm, 2000;Fjeldskaar and
Amantov, 2018;Lindholm et al., 2000;Olesen et al., 2013;Stein et al., 1989).

## 505 7. Conclusions

Analytical modelling of the stress field generated by oblique spreading at the Molloy and Knipovich ridges in the
Fram Strait, suggests that spatial variations in the tectonic stress regime along the Vestnesa Ridge are plausible.
Thus, mid-ocean ridge spreading may be an important factor controlling faulting and seepage distribution in the
region. Other important sources of stress such as gravitational forcing and lithospheric bending, contributing to
the actual state of stress off Svalbard, are not considered in the modelling exercise presented here. Hence, we
cannot quantitatively assess whether ridge push has a dominant effect on seepage activity. However, provided a
certain degree of coupling between crustal and near-surface deformation, it is plausible that stresses from plate
spreading may affect the behaviour of Quaternary faults along the Vestnesa Ridge and exert a certain control on
seepage. Our study supports a tectonic explanation for the observed seepage pattern in the region. The influence
of rifting at the Knipovich Ridge dominantly on the eastern Vestnesa Ridge may be the key for understanding
focusing of present day seepage activity along the ridge. The opening of faults and fractures favourably oriented
with respect to principal stresses combined with a diminished effective stress in a tensile stress regime facilitates
the release of gas from zones of relatively high-pore fluid pressure at the base of the gas hydrate stability zone.
Multiple seepage events along the entire extent of the Vestnesa Ridge, may have been induced by additional
sources of stress likely associated with glacial isostasy. Future reactivation of currently dormant pockmarks or
increase in seepage activity is likely following the gradual westward propagation of the tensile stress zone on the
Vestnesa Ridge as the Eurasian plate drifts towards the south-east. Despite the simplifying assumptions by the
analytical model approach implemented here, this study provides a first assessment of how important
understanding the state of stress is for reconstructing seepage activity along passive margins.

## 526 8. Outlook

The effect of glacial stresses over the fluid flow system off west-Svalbard will be further tested (at least for the
Weichselian period) by implementing Lund et al., models using newly constrained Barents Sea ice-sheet models
(e.g., Patton et al., 2016). Additional sources of stress related to topography/bathymetry should be further
investigated as well to gain a comprehensive assessment of the effect of the total stress field on near-surface fluid
migration in the region.
**Figures**

**Figure 1: (a) International Bathymetry Chart of the Arctic Ocean (IBCAO) showing the geometry of mid-**
**ocean ridges offshore the west-Svalbard margin; (b) High resolution bathymetry along the Vestnesa Ridge**
**(UiT, R/V HH multi-beam system). Seafloor pockmarks are observed along the entire ridge but acoustic**
**flares are restricted to the eastern segment; PKF=Prins Karls Forland; STF=Spitsbergen Transform**

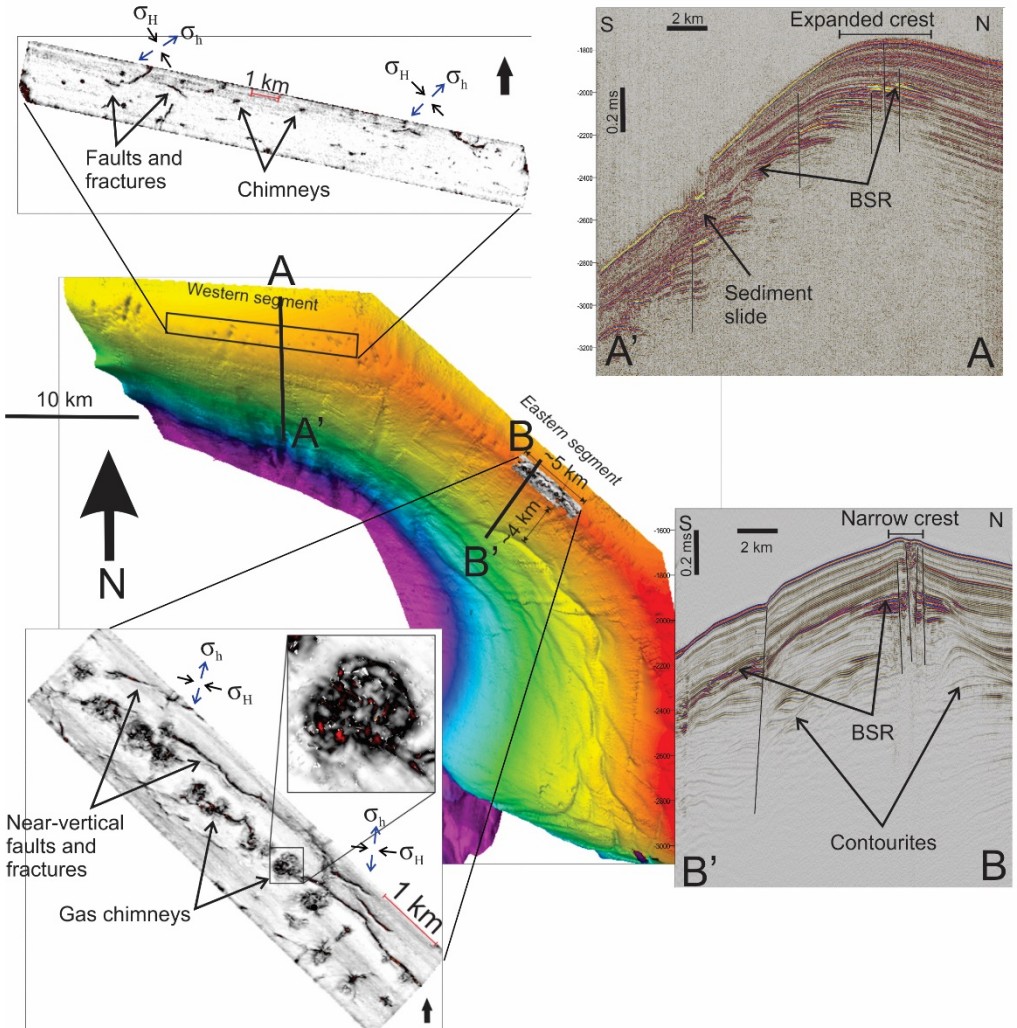


**Figure 2: Composite figure with bathymetry and variance maps from 3D seismic data along the eastern**
**and the western Vestnesa Ridge segments (modified from Plaza-Faverola et al., 2015). The orientation of**
**maximum compressive horizontal stress ($\sigma_H$) and minimum horizontal stress ($\sigma_h$) predicted by the model**
**are projected for comparison with the orientation of fault segments. Notice favourable orientation for**
**opening to fluids on the eastern Vestnesa Ridge segment. Two-2D seismic transects (A-A' - Bünz et al.,**
**2012 and B-B' – Johnson et al., 2015) illustrate the morphological difference of the crest of the Vestnesa**
**Ridge (i.e., narrow vs. extended) believed to be determined by bottom current dominated deposition and**
**erosion (Eiken and Hinz, 1993). BSR=bottom simulating reflector.**


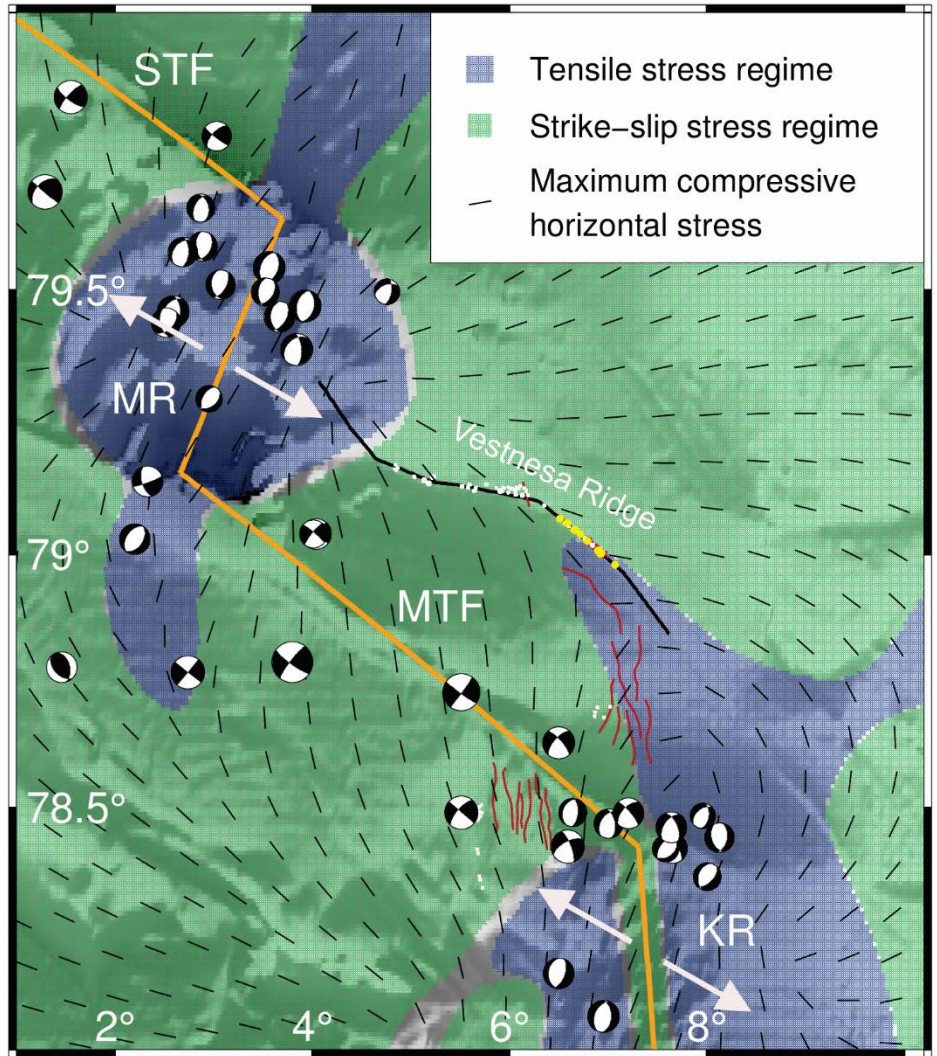


**Figure 3: Modelled upper crustal tectonic stress field (blue – tensile and green - strike-slip regime) and**
**stress orientations, due to oblique spreading at the Molloy Ridge (MR) and the Knipovich Ridge (KR). The**
**outline of a seismic line (Plaza-Faverola et al., 2017) is projected as reference for the crest of the Vestnesa**
**Ridge. Red lines are faults, yellow dots seeps and white circles pockmarks where no acoustic flares have**
**been documented. STF=Spitsbergen Transform Fault; MTF=Molloy Transform Fault. The focal**
**mechanisms are from the ISC Online Bulletin (http://www.isc.ac.uk).**

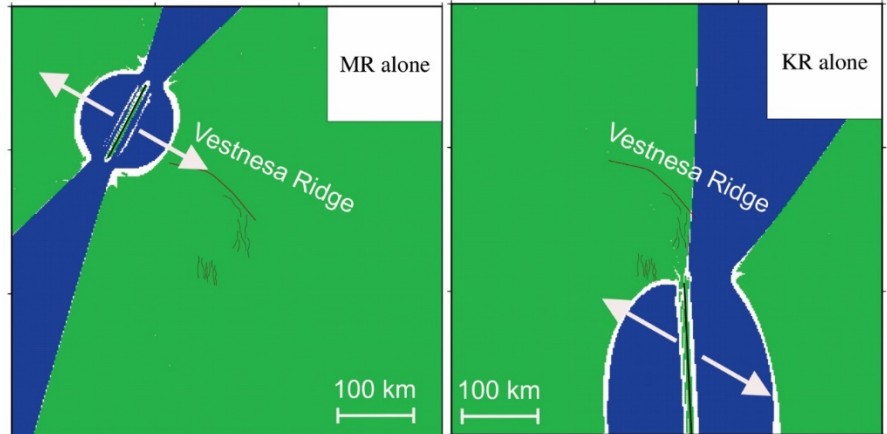


**Figure 4: Stress field resulting from model runs with Molloy Ridge and Knipovich Ridge, respectively:**
**tensile stress field (blue); strike-slip stress field (green).**

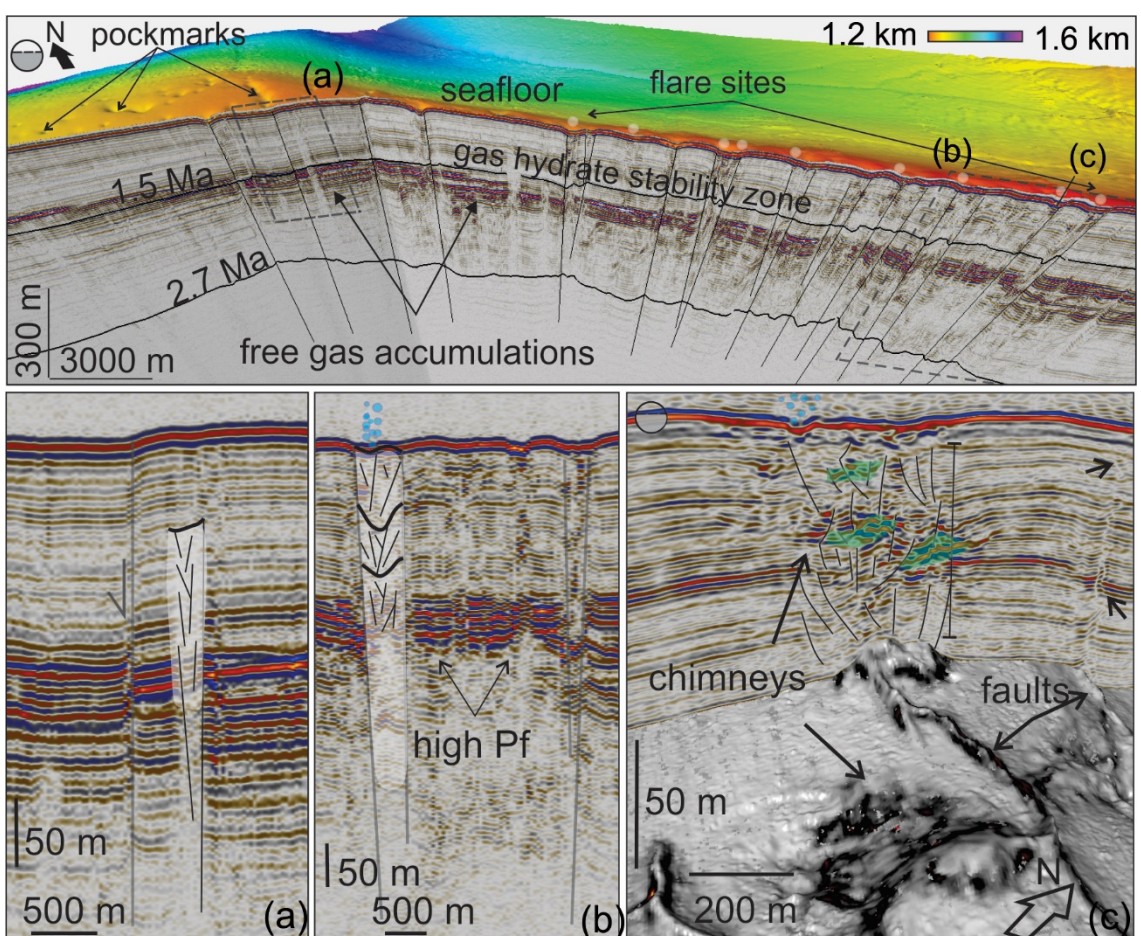


**Figure 5: Integrated seismic and bathymetry image of the gas hydrate system along the Vestnesa Ridge. (a) Outcropping N-S oriented fault located at the transition from the region where acoustic flares have been documented to the region where no flares have been observed; (b) Gas chimneys with associated acoustic flare and inferred high pore-fluid pressure (Pf) zone at the base of the gas hydrate stability zone; (c) Gas chimney associated with faults and faults extending to near-surface strata without being associated with chimneys. The same variance map in figure 2 is projected at the depth where the map was extracted along a surface interpreted on the 3D seismic volume. Green patches represent interpreted zones of buried authigenic carbonate that can activate a self-sealing mechanism leading to hydrofracturing and chimney development.**

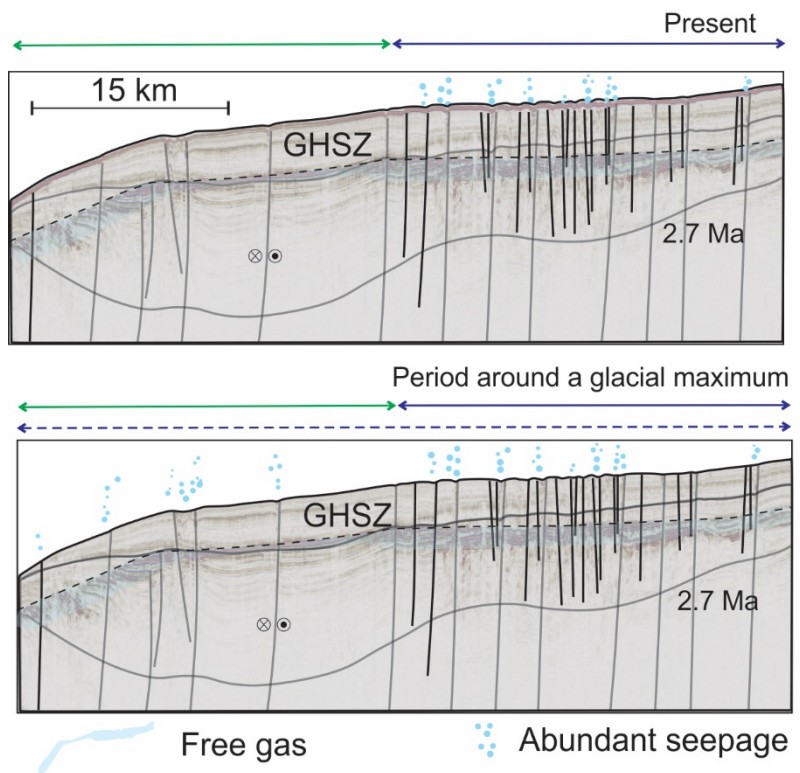

575

**Figure 6: Conceptual model of the evolution of seepage coupled to faulting and spatial variations in the stress regime (tensile=blue; strike-slip=green) along the Vestensa Ridge, offshore the west-Svalbard margin. At present day, tensile stress from mid-ocean ridge spreading (blue solid line) favours seepage exclusively on the eastern segment of the Vestnesa Ridge. Seepage on the western Vestnesa Ridge and other regions may have been induced repeatedly since the onset of glaciations 2.7 Ma ago (Mattingsdal et al., 2014), due to tensional flexural stresses (dashed blue line) in the isostatic forebulge around the time of glacial maximums; GHSZ=gas hydrate stability zone. The dashed black line follows the bottom simulating reflector which represents the base of the GHSZ.**

584

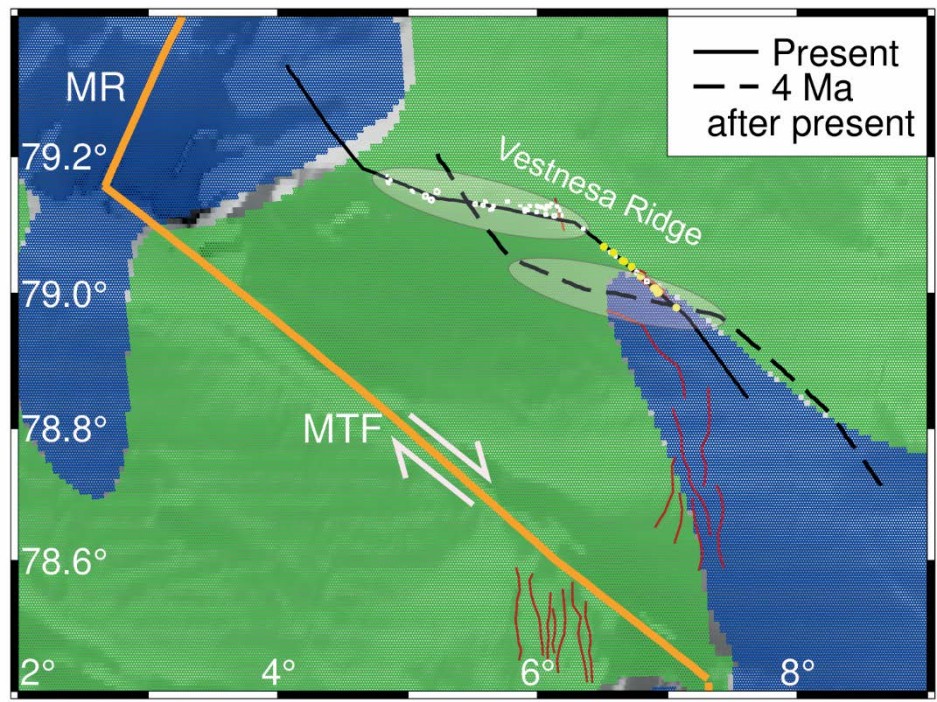

585

**Figure 7: Stress field as in figure 3 showing the location of the Vestnesa Ridge at present and 4 Ma after present time, assuming a constant spreading velocity of 7 mm/yr in the direction N125°E. The same line outline as in figure 3 is used as reference for the crest of the Vestnesa Ridge. Yellow and white dots represent pockmarks with and without documented acoustic flares respectively.**

590

**Appendix A**

**Model description**

593

We use the analytical formulations of Okada (1985) for a finite rectangular dislocation source in elastic homogeneous isotropic half-space (Fig. A.1). The dislocation source can be used to approximate deformation along planar surfaces, such as volcanic dykes (e.g. Wright et al., 2006), sills (e.g. Pedersen and Sigmundsson, 2004), faults (e.g. Massonet et al, 1993) and spreading ridges (e.g. Keiding et al., 2009). More than one dislocation can be combined to obtain more complex geometry of the source or varying deformation along a planar source. The deformation of the source can be defined as either lateral shear (strike-slip for faults), vertical shear (dip-slip at faults) or tensile opening.


The Okada model assumes flat Earth without inhomogeneities. While the flat-earth assumption is usually
adequate for regional studies (e.g. Wolf, 1984), the lateral inhomogeneities can sometimes cause considerable
effect on the deformation field (e.g. Okada, 1985). However, the dislocation model is useful as a first
approximation to the problem.
At mid-ocean ridges, deformation is driven by the continuous spreading caused by gravitational stress due to the
elevation of the ridges, but also basal drag and possibly slab pull. Deformation occurs continuously in the ductile
part of the crust. Meanwhile, elastic strain builds in the upper, brittle part of the crust. To model this setting, the
upper boundary of the dislocation source must be located at the depth of the brittle-ductile transition zone. The
lower boundary of the source is set to some arbitrary large depth to avoid boundary effects.

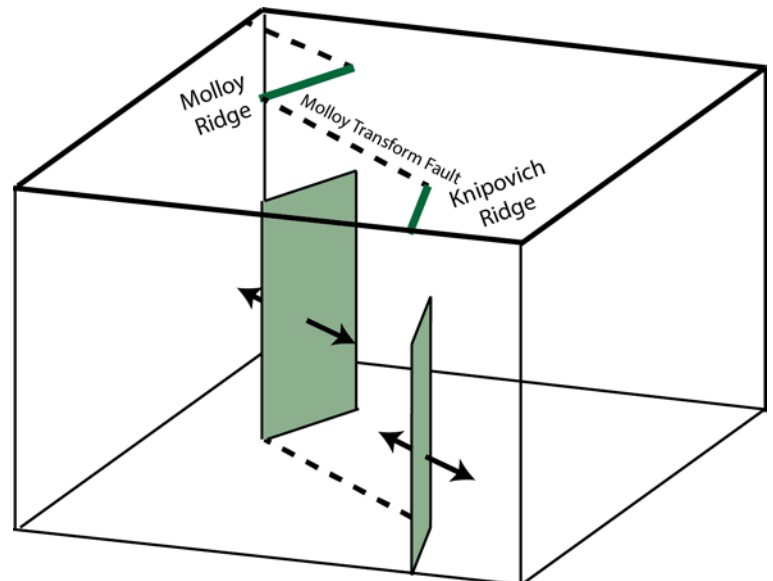

**Fig A.1 Extract of model showing the location of the dislocation sources (light green) for Molloy and**
**Knipovich ridges. Note that the model is an infinite half-space, i.e. it has no lateral or lower boundary.**
The Okada model provides the displacements $u_x$, $u_y$, $u_z$ (or velocities if deformation is time-dependent) at defined
grid points at the surface and subsurface. It also provides strain (or strain rates) defined as:

$$\varepsilon_{ij} = \frac{1}{2}\left(u_{i,j} + u_{j,i}\right)$$


The stress field can then be calculated from the predicted strain rates. In homogeneous isotropic media, stress is
related to strain as:
$$\sigma_{ij} = \lambda\delta_{ij}\varepsilon_{kk} + 2\mu\varepsilon_{ij}$$


where $\delta_{ij}$ is the Kronecker delta, $\lambda$ is Lamé's first parameter, and $\mu$ is the shear modulus. Lamé's first parameter
does not have a physical meaning but is related to the shear modulus and Poisson's ratio (ν) as $\lambda = \frac{2\mu\nu}{1-2\nu}$.

The absolute values of stress are in general difficult to model (e.g. Hergert and Heidbach, 2011), and not possible
with our analytical model. However, the model provides us with the orientations and relative magnitude of the
stresses. That is, we know the relative magnitudes between the vertical stress ($\sigma_v$), maximum horizontal stress
($\sigma_H$) and minimum horizontal stress ($\sigma_h$). From this, the stress regime can be defined as either tensile ($\sigma_v > \sigma_H >$
$\sigma_h$), strike-slip ($\sigma_H > \sigma_v > \sigma_h$) or compressive ($\sigma_H > \sigma_h > \sigma_v$).

**Author contribution**
Andreia Plaza-Faverola conceived the paper idea. She is responsible for seismic data processing and
interpretation. Marie Keiding did the tectonic modelling. The paper is the result of integrated work between both.

**ACKNOWLEDGEMENTS**
This research is part of the Centre for Arctic Gas Hydrate, Environment and Climate (CAGE) supported by the
Research Council of Norway through its Centres of Excellence funding scheme grant No. 223259. Marie Keiding
is supported by the NEONOR2 project at the Geological Survey of Norway. Special thanks to Björn Lund, Peter
Schmidt, Henry Patton, and Alun Hubbard for their interest in the present project and constructive discussions
about isostasy and glacial stresses. We are thankful to various reviewers that have significantly contributed to the
improvement of the manuscript. Seismic data is archived at CAGE – Centre for Arctic Gas Hydrate, Environment
and Climate, Tromsø, Norway and can be made available by contacting APF. Modelled stresses can be made
available by contacting MK.

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
