# Peer review of "CORRELATION BETWEEN TECTONIC STRESS REGIMES AND METHANE SEEPAGE ON THE 2 WEST-SVALBARD MARGIN"

_Solid Earth, 2018_

## Referee Comment (RC1) · Anonymous Referee #1 · 10 May 2018

The paper proposes to correlate the location of active methane seepage zones observed along the west-Svalbard margin to tectonic stresses associated with the Molloy and Knipovich spreading centers of North Atlantic Ridge. They identify two other possible sources of stress, namely gravitational stresses due to topography and flexural stresses due to sediment erosion and deposition.

To my surprise no reference is made to the stress field generated by well documented on going glacial rebound effect.

Authors consider that only spreading centers are relevant and propose to use Okada's elastic solution for dislocations in an infinite elastic space to model the stress gener-

ated by the two spreading centers they are interested in. Interestingly they place the spreading centers below the brittle-ductile transition and assume a 7 mm/y opening rate. Hence, not only they use an elastic solution for analyzing the opening of a dislocation in the ductile part of the lithosphere, but they assume symmetry for the velocity of plates on both sides of the ridge, a feature which ought to be discussed.

Finally they consider that the pore pressure associated with the seepage of methane is larger than the minimum principal stress in the rock formation. But when pore fluid pressure is larger than the minimum principal stress, a hydraulic fracture is formed that keeps propagating till the pressure is released and becomes smaller than the minimum principal stress. This should have been discussed.

I personally completely disagree with authors proposition that the glacial rebound does not affect presently the stress field and is negligible as compared to the effect of the spreading centers. In addition topography effects ae most likely significant an the appropriatness of neglecting them should be demonstrated.

Independently, because of the above mentioned difficulties concerning the proposed model : 1) with using Okada's elastic solution for modeling the stress field generated by a dislocation in a ductile material, 2) by assuming symmetry of plate motions on both sides of the ridge, 3) by considering that hydraulic fractures may remain stable for long durations of time, I cannot accept the paper as is.

I propose a complete revision that will include a discussion showing why all my comments here above are irrelevant.

---

## Short Comment (SC1) · 15 May 2018

Thank you for the comments. We would like to answer/clarify some of the points raised by the reviewer:

1. Modelling approach

The reviewer claims that "no reference is made to the stress field generated by well documented on going glacial rebound effect". This is not correct. We state several times that the study area was most likely affected by glacially induced stresses during the Quaternary (pages 1, 3, 5-6). A whole paragraph of the discussion section is ded-

icated to the discussion of the influence of glacial isostatic adjustment. We reference state-of-the-art work on the stresses generated by glacial isostatic adjustment (Lund & Schmidt, 2011). We suggest that the influence of glacial stresses is probably particularly important to explain seepage activity along the entire Vestnesa Ridge in the past, where seabed pockmarks does not show seepage activity at present.

The reviewer states that we "identify" two other possible sources of stress. This is also incorrect or reflects a misunderstanding. We mention that the study area is affected by other sources of stress (topography, subsurface density contrasts, erosion/deposition, and during the Quaternary also glacially induced flexural stresses). However, we explain that we do this modeling exercise with a clear intention of investigating the stress from tectonic spreading exclusively. It is unfortunate that the reviewer missed this point. This is the essence of the paper. Are there spatial variations in the stress field in the region due to the way the Molloy and the Knipovich ridges are spreading with respect to each other?

The reviewer "disagree with the authors proposition that the glacial rebound does not affect presently the stress field and is negligible as compared to the effect of the spreading centers". While the present study focusses on stress from spreading, we clearly do not claim that other stress sources are negligible. However, models of stress from GIA show that the present-day stress at formerly glaciated margins is small, on the order of 1-2 MPa (Lund et al., 2009; Lund & Schmidt, 2011; van der Wal et al., 2013; Steffen et al., 2006; Steffen et al., 2014). As mentioned, we believe there may have been a more significant effect of glacial stresses on the region in the past. This is work in progress.

2. Ridge asymmetry

The reviewer correctly writes that the Okada model assumes symmetry of the spreading centers. We are aware that some mid-ocean ridges are known to be asymmetric, however, the geometry of the spreading centers in the Fram Strait is not well known and asymmetry has to our knowledge not been documented. Thus, a discussion of

asymmetry can be hardly fit into the scope of our present study. Other studies in the region have also assumed symmetry of the spreading ridges (for example, Johnson et al, 2015). Based on the comment from the reviewer, we will revise the manuscript to describe this assumption explicitly.

The Okada model does, admittedly, involve simplifying assumptions, such as symmetry. However, Árnadóttir et al. (2009) demonstrated that the deformation field from the complex plate boundary in Iceland could be modelled using Okada models. The predicted stress directions from Okada models are in general agreement with other models of plate tectonic forces (for example, Bott, 1991; Gölke & Coblentz, 1996; Fejerskov & Lindholm, 2000; Naliboff et al., 2012). The good agreement between the predicted stress field and the observed focal mechanisms, furthermore, indicate that the model correctly predicts the first order stress field at upper crustal depths. We mention that we do not attempt to analyze the total magnitude of stresses, but relative stress variations in the region caused only by oblique spreading. We plan to test and validate the results of this work in future using other modelling approaches.

3. Stress relief

The reviewer claims that we consider "that hydraulic fractures may remain stable for long durations of time". We do not really go into depletion of pressure following hydraulic fracturing. Here we propose a conceptual model to describe how the formation or opening of already in place faults or new tension fractures may increase secondary permeability and thus lead to seepage. The relevant idea here is that seepage would then occur exclusively in places where the stress regime favors a permeable behavior of faults and fractures. We agree that the formation of hydraulic fractures potentially leads to more complex cycles of pore pressure depletion and build-up that will in turn influence the timing of seepage activity. This is however a local mechanism that makes sense to investigate in the context of detailed imaging of gas chimneys or other near surface fluid flow migrating features (e.g., Hustoft et al., 2009, Hustoft et al., 2010; Gay et al., 2007; Gay et al., 2012). Here we are interested in the regional and more longterm stress regime (e.g., stressing from plate tectonics, glacial isostatic adjustment) and the relation with pore fluid pressure in terms of effective stresses.

Andreia Plaza-Faverola and Marie Keiding

References:

Árnadóttir, T., Lund, B., Jiang, W., Geirsson, H., Björnsson, H., Einarsson, P., Sigurdsson, T., 2009. Glacial rebound and plate spreading: Results from the first countrywide GPS observations in Iceland. Geophysical Journal International, Volume 177, Issue 2, 691–716.

Bott, M. H. P., 1991. Ridge push and associated plate interior stress in normal and hot spot regions. Tectonophysics 200, Issues 1–3, 17-32.

Fejerskov, M. and Lindholm, C., 2000. Crustal stress in and around Norway: an evaluation of stress-generating mechanisms. Geological Society, London, Special Publications, 167, 451-467.

Gay, A., and Berndt, C., 2007, Cessation/reactivation of polygonal faulting and effects on fluid flow in the Vøring Basin, Norwegian Margin: Journal of the Geological Society, v. 164, no. 1, p. 129-141.

Gay, A., Mourgues, R., Berndt, C., Bureau, D., Planke, S., Laurent, D., Gautier, S., Lauer, C., and Loggia, D., 2012, Anatomy of a fluid pipe in the Norway Basin: Initiation, propagation and 3D shape: Marine Geology, v. 332, p. 75-88.

Gölke, M., Coblentz, D., 1996. Origins of the European regional stress field. Tectonophysics 266, 11-24.

Hustoft, S., Bunz, S., Mienert, J., Chand, S., 2009. Gas hydrate reservoir and active methane-venting province in sediments on < 20 Ma young oceanic crust in the Fram Strait, offshore NW-Svalbard. Earth and Planetary Science Letters 284, 12-24.

Hustoft, S., Bünz, S., and Mienert, J., 2010, Three‐dimensional seismic analysis

of the morphology and spatial distribution of chimneys beneath the Nyegga pockmark field, offshore mid‐Norway: Basin Research, v. 22, no. 4, p. 465-480.

Johnson, J. E., Mienert, J., Plaza-Faverola, A., Vadakkepuliyambatta, S., Knies, J., Bünz, S., Andreassen, K., Ferré, B., 2015. Abiotic methane from ultraslow-spreading ridges can charge Arctic gas hydrates. Geology 43 (5), 371-374.

Lund, B., Schmidt, P., 2011. Stress evolution and fault stability at Olkiluoto during the Weichselian glaciation. Posiva Oy working report 2011-14.

Lund, B., Schmidt, P., Hieronymus, C., 2009. Stress evolution and fault stability during the Weichselian glacial cycle. SKB Technical Report TR-09-15.

Naliboff, J. B, Lithgow-Bertelloni, C., Ruff N. de Koker, L.J., 2012. The effects of lithospheric thickness and density structure on Earth‱s stress field. Geophysical Journal International, Volume 188, Issue 1, 1–17.

Steffen, H., Kaufmann, G., Wu, P., 2006. Three-dimensional finite-element modeling of the glacial isostatic adjustment in Fennoscandia. Earth and Planetary Science Letters 250, Issues 1–2, 358-375.

Steffen, H., Kaufmann, G., Lampe, R., 2014. Lithosphere and upper-mantle structure of the southern Baltic Sea estimated from modelling relative sea-level data with glacial isostatic adjustment, Solid Earth, 5 (1), 447-459.

van der Wal, W., Barnhoorn, A., Stocchi, P., Gradmann, S., Wu, P., Drury, M., Vermeersen, B., 2013. Glacial isostatic adjustment model with composite 3-D Earth rheology for Fennoscandia, Geophysical Journal International, 194 (1), 61-77.

---

## Referee Comment (RC2) · Anonymous Referee #1 · 16 May 2018

In lines 110 to 113 of authors paper, it is written: "Because the model only incorporates plate spreading, it is likely that the actual stress field on the west-Svalbard margin differs to some extent from the stress field predicted by our model. However, by excluding all other sources of stress, we are able to investigate the influence of tectonic stress exclusively."

I consider this statement demonstrates an error of judgement: the ongoing methane seepage depends on the coupling between fluid pressure and the presently existing complete stress field, as explained here after.

On line 114, authors state that they use Okada model of dislocations for modeling what

they call tectonic stresses. This assumes elasticity. In elasticity, if four different loading processes are considered, the superposition of all of them at the same time implies that the resulting stress field may be evaluated from the sum of the four stress fields computed independently for each of the loading processes.

Authors have listed as loading mechanisms: A ridge opening, B topography, C effect of sediment erosion-deposition, D flexural stresses due to glaciation.

Hence, according to authors, present stress field result from A+B+C+D. Claiming that it can be investigated by looking at A only, implies that B+C+D are negligible. This requires a demonstration! Nowhere have I seen in the paper computations for B, C, and D.

When I say "no reference is made to well documented on going glacial rebound", this is precisely what I mean. I do not mean authors have not cited previous work, I am saying they have not compared the magnitude of the glacial rebound effect to that of ridge opening at the location of methane seepage.

As a reviewer of a scientific paper, I am careful to check facts, not speculations. I do not consider that authors response to my review do address properly the issue of quantifying effects B, C, and D.

I also do not wish to get involved into endless discussions on whether authors understand what is hydraulic fracturing or not, etc. . ..

I just did what I consider the work of a reviewer should be, i.e. check facts or validity of computations; I will leave the editor in chief decide whether my comments are relevant or not.

I will stop here my time devoted to this paper and do not wish to be further involved in reviews for the journal "Solid Earth". Indeed, I am not interested in discussing opinions. . .my small education just helps me with scientific demonstrations within my very small field of expertise...

---

## Editor Comment (EC1) · F. Rossetti (Editor) · 16 May 2018

As handling editor of this SE Discussion paper, I would remark that the open-access platform of SE should ensure all the different opinions are adequately represented during the editorial workflow, hopefully providing a fair and unbiased review process. Regarding this specific case, I think comments raised by the Referee #1 in his RC1 and re-iterated in RC2 are relevant and should be taken into account and properly discussed by the Authors when preparing a revised version. In particular, I concur with the Referee #1 that a sensitivity analysis of the resulting stress fields is necessary to better support the model results. Federico Rossetti

---

## Author Comment (AC1) · 22 May 2018

Andreia Plaza-Faverola and Marie Keiding

andreia.a.faverola@uit.no

We are thankful for the comments. We acknowledge that we need to add emphasis on some of the key points to ensure the message is conveyed clearly. The comments by reviewer 1 will guide the revision of the manuscript.

Our main goal is to report on a modeling exercise that has a clear intention of investigating the kind of stresses generated exclusively by oblique spreading at both mid-ocean ridges in the Fram Strait (Knipovich and Molloy) and how these stresses correlate with the distribution of seepage activity. While the modeling approach is highly simplified, i.e., based on assumptions of isotropic, homogeneous, elastic rheology as well as flat

earth and spreading symmetry, it represents a first and important step for the investigation of near-surface stresses in this Arctic region. Our next steps will consist in modeling stresses from glacial dynamics to finally integrate different sources of stress in a numerical model of higher complexity.

For improving the manuscript, we intend to:

1. Ensure that we convey clearly that the modeling attempts to investigate stresses exclusively due to the spreading at the Molloy and Knipovich slow spreading ridges (i.e., that other sources of stress are neglected in the model). Hence, the modelled stresses are not to be understood as the actual stress field in the region. In addition, the discussion will address more clearly that there are other potential sources that influence present and past stress fields in the region. We will argue more substantially that regardless the influence of other stress sources, the tensile zone predicted by our models may on itself explain present day seepage distribution. We will also discuss that assuming that seepage occurs where permeability of faults and fractures is favored in a tensile regime, other sources of tensile stress were needed in the past to explain seepage where seabed pockmarks are inactive today.

2. Make a more explicit reference to the sensitivity test we include as supplement. The model we report on here, indicates that there are spatial changes in the stress field caused exclusively by oblique spreading at the mid-ocean ridges which coincide with a change in seepage activity and can be supported by the morphology and distribution of faults. While there is no direct stress observations from bore holes in the region, indirect stress observations from earthquake focal mechanisms support the stress field predicted by the model. The paper includes a sensitivity test showing how the distribution of the predicted tectonic stress field changes while changing key parameters in the model. A tensile zone extending north of the Knipovich ridge seems to be a robust feature of the model. This is a key result because this zone of tensile stress coincides with the zone of active seepage at present day. In addition, the model results support the postulation by Crane et al., 2002 and Vanneste et al., 2005 where outcropping faults

north of the northward termination of the Knipovich ridge indicate northward propagation of the rift system (i.e., this is the region of tensile stresses predicted by the models we present here).

3. Discuss more substantially the validity (or not) of the symmetry assumption in the Okada models. Different studies rely in this assumption of symmetry in this Arctic region based on what can be observed from available magnetic anomaly maps (e.g., Johnson et al., 2015).

4. Extend the discussion about the effect of present-day glacially induced stress on Vestnesa (on the order of 0-2 MPa based on analogy with the models from the Fennoscandia area; by e.g. Lund et al., 2009; Steffen et al., 2006).

Andreia Plaza-Faverola and Marie Keiding

---

## Referee Comment (RC3) · Anonymous Referee #2 · 23 May 2018

General comments. The paper investigates the relationships between tectonic stress generated by spreading along the Molloy and the Knipovich oceanic ridges and the presence of active and extinct seepage along the Vestensa ridge, offshore western Svalbard. The tectonic stresses has been modeled assuming dislocation in an infinite elastic space (Okada's model). The authors find a good correlation between the areal distribution of tensile stress and the occurrence of active seepage, whereas the extinct seeps invariably fall within areas characterized by a strike-slip regime. However, as the authors correctly acknowledge, there is a good but not exact correspondence between active seepage and tensile stress distribution. This analysis has taken into account exclusively the tectonic forces, an approach that has been criticized by Reviewer 1, who

has called upon the role of glacial rebound. In my opinion, the study area is located near active structures, and the contribution of tectonic stress may be worth of investigation. However, there are some important issues that need to be addressed more carefully, as detailed below in the Specific comments. In particular, the description of the structural setting of the Vestensa ridge and the geometrical relationships between the active stress field and the orientation of faults should be more stringent. On the whole, the manuscript is clearly laid out, and has the potential to appeal to a general international audience.

Specific comments. 1. Structural setting of the Vestensa ridge. The structural setting of Vestensa ridge is of crucial importance for understanding active and relict seepage that has localized on this structure. However, description of structural geology is sloppy in many points. To start with, the manuscript misses a discussion regarding the possible genetic relationships between the Vestensa ridge and the Molloy/Knipovich oceanic ridges and their associated transform faults. In addition, origin, age and tectonic structure of the Vestensa ridge have not been discussed. The seismic section illustrated in figure 2 shows the geometry of a gentle anticline. I assume that this anticline corresponds to the Vestensa ridge, yet no location of this seismic section is reported. In addition, the Vestensa ridge shows a marked variation of its trend, with its western sector trending NW and the eastern sector oriented ca. NNW. Does this variation correspond to a difference in structural controls? Panel (b) of Figure 1 should be expanded conveniently to illustrate the location of active and extinct seeps, together with the trace of faults and the anticline axis shown in Figure 2. This would help the reader to visualize better the structural setting of the study area.

2. Fold activity. As far as I can tell, the 'Vestensa' anticline deforms post-1.5 Ma Pleistocene sediments. A central point thus regards the establishment of whether the fold is still active or not. This point may be important in that anticlines are the preferential locus of active seepage because they trap the raising fluids at the fold core. Outer arc (extrados) normal faults may thus provide efficient fluid pathways. Many of the faults

dipping toward the fold core (sketched on Figure 2seismic section) could belong to this category. The amplification of this fold would thus be accommodated by the formation of new faults and/or the opening of existing ones. This possibility could be relevant in case this fold has been controlling active seepage. Again, this calls upon the requirement for a better definition of the structural setting of the Vestensa ridge (point above).

3. Geometric relationships between stress field and pre-existing faults. A interesting point suggested by the modelling results is that existing normal faults could be opened by the operating tensile stress. Normal faults experience sealing-opening cycles that are typically dictated by fluid pressure pulses. On the other hand, this behavior is also controlled by the geometric relationships between the orientation of stress axes and the pre-existing structures. One can note in Figure 3 that active seepage occurs along a NW-trend, whereas inactive seeps occur along a ca. E-W trend. I wonder whether active seepage is depending upon the geometrical relationships between the orientation of regional stress field and the trend of faults. The distinction between active and relict seepage is essentially based on the assumption that a tensile stress regime favors seepage whereas a strike-slip one would not. This reasoning may be not invariably true because strike-slip faults are often steep and connect the subsurface reservoir to the surface, thereby representing efficient fluid pathways. As a matter of fact, there are many examples worldwide where active seepage focuses on both inactive and active strike-slip faults, as well as extensional jogs forming along strike-slip fault systems. In this regard, the manuscript should discuss more deeply why seepage along faults that fall into areas with strike-slip regime is discouraged. Is it because the maximum horizontal stress SH is sub-orthogonal to fault trend? In case the maximum horizontal stress SH is favorably oriented for reactivation, faulting would instead favor fluid upraising. This point could be resolved by showing the orientation of SH and/or Sh axes throughout the study area, together with fault traces on the Vestensa ridge.

4. Earthquake-induced seepage. It is assumed that (line 61) 'Our study is in line

with observations of earthquake-induced seafloor seepage'. However, it should be noted that seepage and/or paroxysmal activity is not necessarily linked to earthquakes, but generally result from the 'normal' evolution of the system. Earthquakes represent obvious external forcing that may occasionally interfere with the system.

Technical corrections. Title of section GEOLOGICAL SETTING OF THE VESTNESA RIDGE SEEPAGE SYSTEM (lines 65) should be numbered as 2 rather than 1. Numbering of following sections should be changed accordingly.

---

## Author Comment (AC2) · 28 May 2018

First, thank you for constructive and kindly exposed comments. These comments will be very helpful to improve the study. We would like to add a note on how we would proceed based on each specific comment:

Comment 1: We agree in how crucial the structural setting at Vestnesa Ridge is to understand seepage activity. The reason why we initially decided not to dedicate a more substantial section to this here, is that the tectonic setting and the description of faults along the ridge is presented in a previous paper (Plaza-Faverola et al., 2015). This 2015 paper actually provides the basis for the modeling study reported here. In that

paper we discuss the correlation of faults with seepage distribution and we postulate for the first time that the morphology of the Vetsnesa Ridge and seepage activity may be controlled by spatial changes in the stress field. The change from the NW-trending western sector to the NNW-trending eastern sector and a possible explanation for this are also presented in the 2015 paper. We see now the need for presenting a more complete description of the tectonic settings and summary of the observations from previous studies.

The location of the seismic section in figure 2 is shown by a black line in figure 1, however, this should have been clearly written out in the caption. Thanks for pointing this out. We will expand panel b of figure 1, as suggested, to support a more detailed description of the structural setting of the Vestnesa ridge.

Comment 2: The Vetsnesa ridge is a contourite drift and its development as an anticline feature is strongly controlled by bottom currents (i.e., it has a sedimentological origin). However, the observation by the reviewer regarding the focusing of gas is of course important to keep in mind for our discussion here. Bünz et al., 2012 discuss how there is a structural control on near-surface gas migration. The gas migrates to the crest of the ridge and gets trapped beneath gas hydrate bearing sediments. Plaza-Faverola et al., 2017 shows that there is a continuous bottom simulating reflector and free gas trapped along the entire ridge extent. The interesting point is that despite the presence of gas all along (e.g., fig 3 in Plaza-Faverola et al., 2017), seepage is only sustained along the easternmost segment. As the reviewer recalls, most likely being related to the capacity of faults to work as fluid pathways. We will make sure that this part of the discussion comes out more substantially in the paper.

Comment 3. The reviewer raises an important question about the geometric relationship between stress field and pre-existing faults. We agree that a discussion about this has to be included in the manuscript. We plan to show the directions of maximum horizontal stress in a figure and discuss how it will affect pre-existing faults. We will clearly describe any known variation in fault orientation along the Vestnesa ridge with respect

to the predicted stress directions. Regarding the type of faults, we agree entirely with the reviewer that seepage is not limited to normal faults. Indeed, we envision that the steep NW-SE oriented faults mapped along the Vestnesa Ridge may have formed in a strike-slip regime, but become permeable during periods of tensile stress. We will revise the manuscript to make sure this important point is clearly communicated.

Comment 4. Thank you for pointing this out. We see that the sentence can be miss-interpreted. With this sentence we intended to emphasize the global relevance of investigating the interplay between regional stress variations in time and fluid discharge. We evoked earthquakes as one example of an external mechanism interacting with the near-surface fluid flow system. We will reformulate.

Marie Keiding and Andreia Plaza-Faverola

---

## Author Comment (AC3) · 15 Jun 2018

We are thankful for the comments. Based on this and comments from referee two, we will restructure and rephrase in places the manuscript to: 1) be sure the main focus of the paper is conveyed clearly: this is a qualitative rather than quantitative study; 2) present a more substantial discussion about the robustness of the resulting tensile zone related to tectonic spreading and the importance of this first approach.

Specific actions on referee 1's comments:

Referee: Authors consider that only spreading centers are relevant and propose to

use Okada's elastic solution for dislocations in an infinite elastic space to model the stress gener-ated by the two spreading centers they are interested in. Interestingly they place the spreading centers below the brittle-ductile transition and assume a 7 mm/y opening rate. Hence, not only they use an elastic solution for analyzing the opening of a dislocation in the ductile part of the lithosphere, but they assume symmetry for the velocity of plates on both sides of the ridge, a feature which ought to be discussed.

Authors: While we agree that asymmetry of the spreading centres in Fram Strait is possible, the presently available magnetic data for the region is not of a quality that justifies such assumption (see, e.g., the NAG-TEC magnetic anomaly map; we note that this question may be resolved after the acquisition of new aeromagnetic data on the west-Svalbard margin by the Geological Survey of Norway).

We again underline the fact that the predicted stresses agree very well with the observed earthquake focal mechanisms indicating that the predicted stress field is - to a first order - a realistic representation of the stresses in the region.

Action: We will modify the discussion to clearly describe the limitations and strengths of implementing Okada's solutions in the investigated setting. We will also describe explicitly that the model assumes symmetry.

Referee: Finally they consider that the pore pressure associated with the seepage of methane is larger than the minimum principal stress in the rock formation. But when pore fluid pressure is larger than the minimum principal stress, a hydraulic fracture is formed that keeps propagating till the pressure is released and becomes smaller than the minimum principal stress. This should have been discussed.

Authors: We propose a conceptual model to describe how the formation or opening of already in place faults or new tension fractures may increase secondary permeability and thus lead to seepage. We agree that the formation of a hydraulic fractures potentially leads to cycles of pore pressure depletion and build-up that will in turn influence the timing of seepage activity.

Action: We will extend on what we already had in the discussion about the cyclic character of the system and how pressure depletion and regional stresses may be interacting.

Referee: I personally completely disagree with authors proposition that the glacial re-bound does not affect presently the stress field and is negligible as compared to the effect of the spreading centers. In addition topography effects ae most likely significant an the appropriatness of neglecting them should be demonstrated. Independently, be-cause of the above mentioned difficulties concerning the proposed model : 1) with using Okada's elastic solution for modeling the stress field generated by a dislocation in a ductile material, 2) by assuming symmetry of plate motions on both sides of the ridge, 3) by considering that hydraulic fractures may remain stable for long durations of time, I cannot accept the paper as is. I propose a complete revision that will include a discussion showing why all my comments here above are irrelevant.

Action: Please see above.

Referee: In lines 110 to 113 of authors paper, it is written: "Because the model only incorporates plate spreading, it is likely that the actual stress field on the west-Svalbard margin differs to some extent from the stress field predicted by our model. However, by excluding all other sources of stress, we are able to investigate the influence of tectonic stress exclusively." I consider this statement demonstrates an error of judgement: the ongoing methane seepage depends on the coupling between fluid pressure and the presently existing complete stress field, as explained here after.

On line 114, authors state that they use Okada model of dislocations for modeling what they call tectonic stresses. This assumes elasticity. In elasticity, if four different loading processes are considered, the superposition of all of them at the same time implies that the resulting stress field may be evaluated from the sum of the four stress fields computed independently for each of the loading processes. Authors have listed as loading mechanisms: A ridge opening, B topography, C effect of sediment erosiondeposition, D flexural stresses due to glaciation. Hence, according to authors, present stress field result from A+B+C+D. Claiming that it can be investigated by looking at A only, implies that B+C+D are negligible. This requires a demonstration! Nowhere have I seen in the paper computations for B, C, and D.

Authors: As previously described, the purpose of the present paper is to report on a modeling exercise with the intention of investigating the kind of stresses (in a qualitative way) generated exclusively by oblique spreading at the mid-ocean ridges in the Fram Strait (Knipovich and Molloy) and how these stresses (alone) seemingly correlate with the distribution of seepage activity.

As the referee recalls from the main text, seepage depends on the coupling between fluid pressure and the present stress field, which is most likely a result of a number of different sources. In this study, we did not attempt to model the entire stress field, rather, we attempt to investigate the influence of plate spreading by considering this source alone. Whether plate spreading is the dominant source of stress or not, will be further investigated by establishing a numerical model of higher complexity that includes additional sources like stresses related to isostatic rebound (works in progress) and topography (what the referee terms as B, C etc.).

While the modeling approach here is simplified, i.e., based on assumptions of isotropic, homogeneous, elastic rheology as well as flat earth and spreading symmetry, it represents a first and important step for the investigation of near-surface stresses in this Arctic region (the study area is adjacent and hence highly vulnerable to mid-ocean ridge stresses, probably more than anything else).

Action: We will modify the paper accordingly to make clear that the study intends to report on an important qualitative observation that forms the basis for understanding the interaction between regional processes and near-surface fluid dynamics in Arctic settings. We will elaborate on the arguments for considering the tectonic stresses to be dominant in this study area.

Referee: When I say "no reference is made to well documented on going glacial rebound", this is precisely what I mean. I do not mean authors have not cited previous work, I am saying they have not compared the magnitude of the glacial rebound effect to that of ridge opening at the location of methane seepage. As a reviewer of a scientific paper, I am careful to check facts, not speculations. I do not consider that authors response to my review do address properly the issue of quantifying effects B, C, and D.

Authors: We realize that we misunderstood the phrasing here. The comment regards the same issue as above, i.e., that we do not attempt to quantify all possible sources of stress (A+B+C etc). We can, however, include a more qualitative discussion about the possible magnitude of stress from glacial isostatic rebound (based on analogy with the models from the Fennoscandia area; by e.g. Lund et al., 2009; Steffen et al., 2006).

Referee: I also do not wish to get involved into endless discussions on whether authors understand what is hydraulic fracturing or not, etc: : :. I just did what I consider the work of a reviewer should be, i.e. check facts or validity of computations; I will leave the editor in chief decide whether my comments are relevant or not. I will stop here my time devoted to this paper and do not wish to be further involved in reviews for the journal "Solid Earth". Indeed, I am not interested in discussing opinions: : :my small education just helps me with scientific demonstrations within my very small field of expertise... C2

Authors: We thank the reviewer for devoting time to review the paper and apologize for any misunderstandings.

Andreia Plaza-Faverola and Marie Keiding

---

## Author Response (AR1)

Dear editor,

Please see below the point by point work based on the 2 referees comments. Major changes include enlargement of the section about tectonic and stratigraphic setting to address main concerns by referee 2; re-structuring of the sections to include a "Results and discussion" section divided into 4 subsections to discuss more focused and substantially the modeling results (including limitations of the assumptions), correlation of modeled stresses with structural features, seepage evolution coupled to tectonic stress variations, and the potential effect of additional stress sources (e.g., from glacial isostasy) mainly on past seepage. We included results on the orientation of the principal modelled stress and discuss the implications of favorably oriented stresses with respect to existing faults in terms of opening for fluids. We added one figure to support the description of the faults and correlation between fault orientation and orientation of modelled tectonic stresses. The figure showing the modeled stresses was updated with the projection of the stress vectors.

We believe the manuscript has been considerably improved we look forward to yours and the referees opinion.

Point by point response to referee 1:

-Authors consider that only spreading centers are relevant and propose to use Okada'selastic solution for dislocations in an infinite elastic space to model the stress gener-ated by the two spreading centers they are interested in. Interestingly they place the spreading centers below the brittle-ductile transition and assume a 7 mm/y opening rate. Hence, not only they use an elastic solution for analyzing the opening of a dislocation in the ductile part of the lithosphere, but they assume symmetry for the velocity of plates on both sides of the ridge, a feature which ought to be discussed.

We modified the text in places to clearly describe the limitations and strengths of implementing the Okada's solutions and provide arguments for why despite the limitations of the model simplicity and assumptions, the results are a realistic first order representation of tectonic stresses in the region. We extended on the symmetry assumption and indicated that this question may be better addressed after the acquisition of new aeromagnetic data on the west-Svalbard margin.

-Finally they consider that the pore pressure associated with the seepage of methane is larger than the minimum principal stress in the rock formation. But when pore fluid pressure is larger than the minimum principal stress, a hydraulic fracture is formed that keeps propagating till the pressure is released and becomes smaller than the minimum principal stress. This should have been discussed.

We restructured the discussion section. We included some statements indicating that mechanisms as hydro fracturing are indeed important modulators of the pore fluid pressure and dynamic behavior of the system. A couple of new references have been added to the discussion.

-I personally completely disagree with authors proposition that the glacial rebound does not affect presently the stress field and is negligible as compared to the effect of the spreading centers. In addition topography effects ae most likely significant an the appropriatness of neglecting them should be demonstrated. Independently, because of the above mentioned difficulties concerning the proposed model : 1) with using Okada's elastic solution for modeling the stress field generated by a dislocation in a ductile material, 2) by assuming symmetry of plate motions on both sides of the ridge, 3) by considering that hydraulic fractures may remain stable for long durations of time, I cannot accept the paper as is. I propose a complete revision that will include a discussion showing why all my comments here above are irrelevant.

Regarding point one and two, we emphasize that the Okada model does, admittedly, involve simplifying assumptions that may not be necessarily perfectly fitting the structural setting in the region. We argue however that the model approach has been implemented for other margins where GPS has been used to validate the results. For example, Árnadóttir et al. (2009). The predicted stress directions from Okada models are in general agreement with other models of plate tectonic forces (for example, Bott, 1991; Gölke & Coblentz, 1996; Fejerskov & Lindholm, 2000; Naliboff et al., 2012). The good agreement between the predicted stress field and the observed focal mechanisms, furthermore, indicate that the model correctly predicts the first order stress field at upper crustal depths (as mentioned above). All these is conveyed in the main text.

For 2 and 3 please see above.

-In lines 110 to 113 of authors paper, it is written: "Because the model only incorporates plate spreading, it is likely that the actual stress field on the west-Svalbard margin differs to some extent from the stress field predicted by our model. However, by excluding all other sources of stress, we are able to investigate the influence of tectonic stress exclusively." I consider this statement demonstrates an error of judgement: the ongoing methane seepage depends on the coupling between fluid pressure and the presently existing complete stress field, as explained here after.

On line 114, authors state that they use Okada model of dislocations for modeling what they call tectonic stresses. This assumes elasticity. In elasticity, if four different loading processes are considered, the superposition of all of them at the same time implies that the resulting stress field may be evaluated from the sum of the four stress fields computed independently for each of the loading processes. Authors have listed as loading mechanisms: A ridge opening, B topography, C effect of sediment erosion-deposition, D flexural stresses due to glaciation. Hence, according to authors, present stress field result from A+B+C+D. Claiming that it can be investigated by looking at A only, implies that B+C+D are negligible. This requires a demonstration! Nowhere have I seen in the paper computations for B, C, and D.

We modified the paper in several places to make clear that the study intends to report on an important qualitative observation that forms the basis for understanding the interaction between regional processes and near-surface fluid dynamics in Arctic settings. We hope that it appears clear now that the modelled stresses cannot be considered a quantitative representations of total stresses in the region because we only investigate the kind of stresses potentially generated by oblique spreading and how these would affect existing faults and associated fluid migration. We elaborated on the arguments for considering the tectonic stresses to be dominant in this study area. We also added arguments for why glacial isostasy at present is not likely to be more dominant than the spreading stress but that it is likely that in the past these stresses were more significant in the region (providing an additional source of stress for explaining seepage in more extended areas along the Vestnesa Ridge).

-When I say "no reference is made to well documented on going glacial rebound", this is precisely what I mean. I do not mean authors have not cited previous work, I am saying they have not compared the magnitude of the glacial rebound effect to that of ridge opening at the location of methane seepage. As a reviewer of a scientific paper, I am careful to check facts, not speculations. I do not consider that authors response to my review do address properly the issue of quantifying effects B, C, and D.

Please see above

-I also do not wish to get involved into endless discussions on whether authors understand what is hydraulic fracturing or not, etc: : :.I just did what I consider the work of a reviewer should be, i.e. check facts or validity ofcomputations; I will leave the editor in chief decide whether my comments are relevantor not.I will stop here my time devoted to this paper and do not wish to be further involved in reviews for the journal "Solid Earth". Indeed, I am not interested in discussingopinions: : :my small education just helps me with scientific demonstrations within my very small field of expertise...

We thank the reviewer for devoting time to review the paper and apologize for any misunderstandings.

Referee 2:

1. Structural setting of the Vestensa ridge. The structural setting of Vestensa ridge is of crucial importance for understanding active and relict seepage that has localized on this structure. However, description of structural geology is sloppy in many points. To start with, the manuscript misses a discussion regarding the possible genetic relationships between the Vestensa ridge and the Molloy/Knipovich oceanic ridges and their associated transform faults. In addition, origin, age and tectonic structure of the Vestensa ridge have not been discussed. The seismic section illustrated in figure 2 shows the geometry of a gentle anticline. I assume that this anticline corresponds to the Vestensa ridge, yet no location of this seismic section is reported. In addition, the Vestensa ridge shows a marked variation of its trend, with its western sector trending NW and the eastern sector oriented ca. NNW. Does this variation correspond to a difference in structural controls? Panel (b) of Figure 1 should be expanded conveniently to illustrate the location of active and extinct seeps, together with the trace of faults and the anticline axis shown in Figure 2.

We restructured section 2 to add substance to the description of the structural setting along the Vestensa Ridge,. We added a figure (new figure 2) to insert the main observations and descriptions from Plaza-Faverola et al., 2015 into this paper. We also use this figure to project the modeled orientation of stresses on imaged faults to analyze favorable orientation of principal stresses for creating opening along the structures (following the suggestion by referee 2).

2. Fold activity. As far as I can tell, the 'Vestensa' anticline deforms post-1.5 Ma Pleistocene sediments. A central point thus regards the establishment of whether the fold is still active or not. This point may be important in that anticlines are the preferential locus of active seepage because they trap the raising fluids at the fold core. Outer arc (extrados) normal faults may thus provide efficient fluid pathways. Many of the faults dipping toward the fold core (sketched on Figure 2seismic section) could belong to this category. The amplification of this fold would thus be accommodated by the formation of new faults and/or the opening of existing ones. This possibility could be relevant in case this fold has been controlling active seepage. Again, this calls upon the requirement for a better definition of the structural setting of the Vestensa ridge (pointabove).

As part of the restructuring of section 2, we also elaborated/clarified that the Vetsnesa ridge is a contourite drift and its development as an anticline feature is strongly controlled by bottom currents (i.e., it has a sedimentological origin). We indicated relevant references. We also put more emphasis on the fact that even if the ridge is not a structural anticline its crest is the focus of fluids migrating toward the highest point of the ridge. We inserted in figure 2, insets from 2 seismic crosslines from Bünz et al and Johnson et al showing the variation on the morphology of the ridge on the western and eastern segments respectively.

3. Geometric relationships between stress field and pre-existing faults. A interesting point suggested by the modelling results is that existing normal faults could be opened by the operating tensile stress. Normal faults experience sealing-opening cycles that are typically dictated by fluid pressure pulses. On the other hand, this behavior is also controlled by the geometric relationships between the orientation of stress axes and the pre-existing structures. One can note in Figure 3 that active seepage occurs along a NW-trend, whereas inactive seeps occur along a ca. E-W trend. I wonder whether active seepage is depending upon the geometrical relationships between the orientation of regional stress field and the trend of faults. The distinction between active and relict seepage is essentially based on the assumption that a tensile stress regime favors seepage whereas a strike-slip one would not. This reasoning may be not invariably true because strike-slip faults are often steep and connect the subsurface reservoir to the surface, thereby representing efficient fluid pathways. As a matter of fact, there are many examples worldwide where active seepage focuses on both inactive and active strike-slip faults, as well as extensional jogs forming along strike-slip fault systems. In this regard, the manuscript should discuss more deeply why seepage along faults that fall into areas with strike-slip regime is discouraged. Is it because the maximum horizontal stress SH is sub-orthogonal to fault trend? In case the maximum horizontal stress SH is favorably oriented for reactivation, faulting would instead favor fluid upraising. This point could be resolved by showing the orientation of SH and/or Sh axes throughout the study area, together with fault traces on the Vestensa ridge.

We followed the reviewer suggestion (as mentioned above) and included in the new figure 2 the projection of the stresses from the modeling figure which now includes the vectors from maximum horizontal compressive stresses as well in the entire modeling area. The geometric relationship between the modeled stresses and preexisting faults should appear clearer now. We clarify that a favorable orientation of the principal stresses can indeed favor opening of faults in the strike slip regime as well. The discussion about seepage coupled to stress field variations and pore fluid pressure interactions has been substantially improved.

4. Earthquake-induced seepage. It is assumed that (line 61) 'Our study is in line with observations of earthquake-induced seafloor seepage'. However, it should be noted that seepage and/or paroxysmal activity is not necessarily linked to earthquakes, but generally result from the 'normal' evolution of the system. Earthquakes represent obvious external forcing that may occasionally interfere with the system.

This sentence and paragraph in the introduction has been reformulated.

Sincerely,

Andreia Plaza-Faverola and Marie Keiding

[revised manuscript text omitted]

---

## Author Response (AR2)

Dear Editor,

We hereby submit the amended version of the manuscript. We have done some significant changes in the intro, results and discussion section. These changes include addition of a sub-section (5.3) to discuss the potential significance of ridge push stresses on fluid dynamics in the context of additional factors likely contributing to the total state of stress in the region. As explained before, the study does not aim at quantifying the different factors causing regional stresses but at evaluating the significance of tectonic stresses on seepage activity. We have thus amended the text in the abstract, the intro, the results and discussion and the conclusion sections to clarify that we argue for a significant effect of tectonic stresses on fluid dynamic without neglecting the significance of other potential sources of stress in the region. We hope this address the major concern by referee one (i.e., that it cannot be concluded that tectonic stresses have a dominant effect on fluid migration in the region before quantifying all potential sources of stress). In addition, we included a short outlook section (7) to indicate important to work that remains to be done and that is planned to further assess the effect of stresses on fluid dynamics in the region.

Please see below a point by point response to the referee comments and the marked-up version of the manuscript.

Sincerely,

Andreia Plaza-Faverola and Marie Keiding

*As pointed out in my review of the original version of this paper, authors rightfully point out that the local stress field, where seepage occurs, may results from four different loading processes:*

*A: gravitational effects through topography effects and density contrasts;*

*B: Flexural effects due to sediment erosion and deposition;*

*C: Glacially flexural effects;*

*D: tectonic stresses resulting from the Vestnesa Ridge.*

*Authors claim that by investigating solely the ridge push effect they can demonstrate that it is the most significant effect that controls methane production. In order to undertake this demonstration they propose to use Okada 's elastic solution that addresses stresses caused by a dislocation in an homogeneous infinite plate.*

**Reply: we do not claim that mid-ocean ridge stress has the most significant effect on seepage; we hypothesize that mid-ocean ridge stress has a significant effect (possibly a dominant effect) on seepage (specifically on the Vestnesa sedimentary ridge; we are not hypothesizing that it is necessarily the case for the entire west Svalbard margin). Additional sources of stress are listed in the paper and it is clearly pointed out that we do not aim at quantifying all the sources of stress in the region.**

Action: We rephrased the text in places to make even more emphatic that we are postulating tectonic stress as a significant factor affecting seepage activity, without disregarding that other sources of stress can as well have a significant effect on fluid migration in the region. We have added a new sub-section (5.3) in the results and discussion section to discuss explicitly other potential sources of stress in comparison to the tectonic stresses.

*I pointed out in my review that, in elastic theory, when various loading processes are applied on the same volume of an elastic material, the resulting stress fields is the sum of that caused by each of the various loading processes applied separately. In the present case, authors must demonstrate that effects of A+B+C are negligible as compared to that of D, their push ridge model, if they wish to conclude that only D is of import. But they fail to do so.*

**Reply: we do not conclude that only the mid-ocean ridge spreading related stresses are of importance.**

Action: please see previous comment.

*Interestingly, authors mention in their introduction that Wallman et al. (2018) conclude that hydrate dissociation off Svalbard is induced by isostatic glacial rebound. So clearly not everybody agrees with their proposition that glacial rebound is negligible.*

**Reply: the study by Wallmann et al., is used for presenting the latest hypothesis for explaining seepage in a region restricted to the shelf break: that gas hydrates dissociate due to post glacial isostatic rebound. This hypothesis first, is relevant only to this region close to the shelf break where the gas hydrate stability zone pinches out and small changes in pressure and temperature easily bring hydrates out of stability (see for example the analyses of pressure changes affecting the gas hydrate reservoir along the Vestnesa Ridge by Plaza-Faverola et al., 2017). It is not valid for > 1000 m water depth, towards the Vestnesa Ridge, where the gas hydrates are perennially stable. Second, the hypothesis does not discuss horizontal stresses, it claims decrease in the hydrostatic pressure is the cause of hydrate dissociation. Third, gas hydrates has not been found in this seepage region.**

Action: we have rephrased the text where we used the reference Wallmann et al., 2018 to emphasize that glacial stresses are actually an additional explanation for sustaining present day seepage at the shelf-break region; something that Wallmann et al. actually disregard.

*Before addressing the stress analysis proposed by authors for the ridge push effect, I will point out that topography effects at a depth of 1200 m is of particularly import given the bathymetry shown on figure 1 , which indicates a slope extending from -1 km to -3km below sea level. Clearly the stress field close to this free surface depends strongly on topography and its effect must be analyzed properly before claiming it is negligible.*

**Reply: we agree that a gravitational component in the faults at the flanks of the Vestnesa basin and on the crest of Vestnesa Ridge is probable. We do indicate this. Nevertheless, the tectonic origin of the faults is put forward based on previous studies observations and our detailed seismic imaging of the faults suggesting that the Knipovich ridge rift system is propagating northward (see Crane et al., 2001; Vanneste et al., 2005; Hustoft et al., 2009; Plaza-Faverola et al., 2015).**

Action: this is now more explicitly discussed in the new discussion section. Please see action for the first comment.

*In Appendix A, Okada's dislocation is embedded in the ductile part of the crust. This is precisely contradictory with the elastic hypothesis of Okada's solution and renders the stress analysis erroneous. In addition, it is stated, p 4 - second paragraph, that the total sedimentary thickness is larger 5 km. Elastic characteristics of sediments differ strongly from that of basement crystalline rocks so that the homogeneous semi-infinite space analysis does not apply for the elastic part of the crust.*

**Reply: The Okada solution has previously been used to correctly model the stresses in the elastic crust by simulating the continuous deformation below the brittle-ductile transformation zone (Arnadotttir et al, 2009). We again point out that the modelled stresses agree very well with stress indications from earthquake focal mechanisms in the region. The reviewer points out that the elastic parameters of sediments vary from the elastic parameters of the crystalline rocks, which we acknowledge. Differences in elastic parameters will result in different magnitudes of stress, but not affect the directions and relative stress magnitudes presented in this study.**

Action: we added text and rephrased in places to better describe the simplifying assumptions and also point out that this study is a first attempt to assess the influence of tectonic stress,

*In conclusion, first the stress field analysis of the ridge push effect is erroneous, second the proposition that topography effects are negligible has not been demonstrated, third results from Wallman et al. show that glacial rebound is an important factor controlling methane production off Svalbard.*

**Reply: See comments above.**

My final recommendation is not to publish this paper.

[revised manuscript text omitted]

Plaza-Faverola et al., (2015) documented seismic differences in the orientation and type of faulting along the ridge and showed a link between the distribution of gas chimneys and faults. They postulated that the faults have a tectonic origin and that spatial and temporal tectonic stress variations may have a long-term effect on the spatial distribution of fault-related gas migration and seepage evolution. In general, the total state of stress at passive continental margins is the result of diverse factors including bathymetry and subsurface density contrasts, subsidence due to sedimentary loading and lithospheric cooling, in addition to ridge-push (e.g., Turcotte et al., 1977). The state of stress at formerly glaciated margins (i.e., such as the west-Svalbard margin) has in addition the effect of flexural bending succeeding ice-sheet advances and retreats (e.g., Fjeldskaar and Amantov, 2018; Grunnaleite et al., 2009; Patton 
[revised manuscript text omitted]

The topography effect on the regional stress regime appears likely along the south-eastern flank of the Vestnesa
basin were the bathymetry deepens from 1200-1600 m along the crest of the Vestnesa Ridge to Ca. 2000 m near
the Molloy Transform Fault (Fig. 1). Small-scale slumps at the slope (Fig 1, 2) may be evidence of gravitational
forcing. Gravitational stress would induce tensile horizontal stress perpendicular to the crest of the Vestnesa ridge.
Thus, faulting and opening would be induced on NW-SE trending faults on the eastern Vestnesa Ridge and WNW-
ESE trending faults on the western Vestnesa Ridge. While gravitational stress may influence present day seepage along the Vestnesa Ridge, it offers no explanation as to why active seepage occurs exclusively on the eastern
Vestnesa Ridge.
The effect of lithospheric bending on the total state of stress in the region remains poorly investigated. It is believed
that the Vestnesa sedimentary Ridge sits over relatively young oceanic crust, < 19 Ma old (Eiken and Hinz, 1993;
Hustoft, 2009). The oceanic-continental transition is not well constrained but its inferred location crosses the
Vestnesa Ridge at its easternmost end (Engen et al., 2008; Hustoft, 2009; Plaza-Faverola et al., 2015). This is a
zone prone to flexural subsidence due to cooling during the evolution of the margin and the oceanic crust may have
experienced syn-sedimentary subsidence nearby the oceanic-continental transition, as suggested for Atlantic
passive margins (Turcotte et al., 1977). However, only N-S oriented faults would be consistent with a deformation
incited by density contrast related flexural bending (i.e., regional structural horses and grabens are N-S trending,
reflecting the direction of major rift systems during basin evolution (Faleide et al., 1991; Faleide et al., 1996).The
NW-SE to E-W oriented faults on the Vestnesa Ridge point towards a different origin.
Lithospheric bending is also associated with Quaternary isostasy during glacial and interglacial periods. Glacial
isostasy results in significant stresses associated with subsidence and uplift of the crust as the ice-sheet advanced
or retreated. Present uplift rates are stronger at the core of the ice-sheets where the ice thickness was at the maximum
(e.g., ~700 m accumulated subsidence has been estimated for the last million year at formerly glaciated basins in
the Barents Sea; (Fjeldskaar and Amantov, 2018). Subsidence and uplift rates decrease towards the former ice-
sheet margin. Indeed, modelled present day uplift rates at the periphery of the Barents sea ice-sheet ranges from 0
to -1 mm/a, depending on the ice-sheet model used in the calculation (Auriac et al., 2016) compared to an uplift
rate of up to 9 mm/a at the centre (i.e., maximum thickness zone) of the ice sheet (Auriac et al., 2016; Patton et al.,
2016). Consistently, modelled glacial stresses induced by the Fennoscandian ice sheet on the mid-Norwegian
margin are close to zero at present day (Lund et al., 2009; Steffen et al., 2006). The Vestnesa Ridge is located ~60
km from the shelf break (Fig. 1). It is actually closer to the Molloy mid-ocean ridge than to the shelf break. It is
likely that the present day effect of glacial stresses on seepage activity in the region is more important towards the
seep sites on the shelf break (i.e., PKF region; Fig. 1) than along the Vestensa Ridge. Wallmann et al., (2018)
postulated that post glacial uplift lead to gas hydrate dissociation and associated seepage on the shelf break. Since
no gas hydrates have been found despite deep drilling (Riedel et al., 2018), the influence of glacial stresses provides
an alternative and previously not contemplated explanation for seepage in this area. It is also likely that glacial stresses as far off as the Vestnesa Ridge had a more significant effect in the past, as further discussed in section
4.5.
Since we focused exclusively on modelling the type of stresses potentially generated by oblique spreading at the
Molloy and Knipovich ridges, and we have so far disregarded any other source of stress, the modelled stress field
in this study cannot be understood as a representation of the total state of stress in the region. However, the striking
correlation between the predicted tensile stress regime with favourably oriented faults, gas chimneys and current
seepage on the eastern segment of the Vestnesa Ridge, suggests that tectonic stresses resulting from oblique
spreading at the Molloy and the Knipovich ridges have the potential to influence near-surface sedimentary
deformation and fluid dynamics in the study area. Tectonic processes at plate margins have a major influence on
regional stress patterns (Heidbach et al., 2010). Given the proximity to the Molloy and Knipovich Ridges, and
without neglecting that other sources of stress can have as well a significant effect on shallow fluid dynamics, we
argue that tectonic stress (ridge push) is an important factor, perhaps even a dominant factor, modulating seepage
activity along the Vestnesa Ridge.
Other stress sources of importance in the region may be gravitational stresses due to bathymetry/topography and
subsurface density contrasts and flexural stresses. During the Quaternary, the west Svalbard margin has
furthermore been affected by glacially induced flexural stresses due to the glaciations (e.g., Fjeldskaar and
Amantov, 2017; Patton et al., 2016). (Auriac et al., 2016; Eiken and Hinz, 1993; Engen et al., 2008; Faleide et al.,
1991; Faleide et al., 1996; Fjeldskaar and Amantov, 2018; Hustoft, 2009; Patton et al., 2016; Pedersen et al., 2010;
Plaza-Faverola et al., 2015)Models of stresses induced by the Fennoscandian ice sheet (Lund et al., 2009; Steffen
et al., 2006) close to zero at presentday.T.
**5.4 PRESENT DAY SEEPAGE COUPLED TO STRESS CYCLING**
Based on the correlation between tectonic stress regimes and seepage patterns, we postulate that current seepage at
the eastern Vestnesa Ridge segment is favoured by the opening of pre-existing faults in a tensile stress regime (Fig.
2, 3b). Depending on the tectonic regime, permeability through faults and fractures may be enhanced or inhibited
(e.g., Faulkner et al., 2010; Hillis, 2001; Sibson, 1994). Thus, spatial and temporal variations in the tectonic stress
regime may control the transient release of gas from the seafloor over geological time as documented, for example,
for $CO_2$ analogues in the Colorado Plateau (e.g., Jung et al., 2014).

[revised manuscript text omitted]

Fram Strait, suggests that tectonic forcing may be an important factor controlling faulting and seepage distribution along the Vestnesa Ridge, off the west-Svalbard margin. Other important sources of stress such as bathymetry and lithospheric bending, contributing to the actual state of stress off Svalbard, are not considered in the modelling exercise presented here; thus, we cannot quantitatively assess whether ridge push has a dominant effect on seepage activity. However, our analysis of how stress from plate spreading may affect mapped faults along the Vestnesa

Ridge suggests that a spatial variation in the tectonic stress regime favours fluid migration through faults on the eastern Vestnesa Ridge where active seepage occurs. We suggest 
[revised manuscript text omitted]
Früh-Green, G.L., Jorgensen, S.L., 2010. Discovery of a black smoker vent field and vent fauna at the Arctic Mid-
Ocean Ridge. Nature Communications 1, 126.
Petersen, C.J., Bünz, S., Hustoft, S., Mienert, J., Klaeschen, D., 2010. High-resolution P-Cable 3D seismic imaging
of gas chimney structures in gas hydrated sediments of an Arctic sediment drift. Marine and Petroleum Geology
doi: 10.1016/j.marpetgeo.2010.06.006, 1-14.
Planke, S., Eriksen, F.N., Berndt, C., Mienert, J., Masson, D., 2009. P-Cable high-resolution seismic.
Oceanography 22, 85.
Plaza-Faverola, A., Bünz, S., Mienert, J., 2011. Repeated fluid expulsion through sub-seabed chimneys offshore
Norway in response to glacial cycles. Earth and Planetary Science Letters 305, 297-308.
Plaza-Faverola, A., Bünz, S., Johnson, J.E., Chand, S., Knies, J., Mienert, J., Franek, P., 2015. Role of tectonic
stress in seepage evolution along the gas hydrate-charged Vestnesa Ridge, Fram Strait. Geophysical Research
Letters 42, 733-742.
Plaza-Faverola, A., Henrys, S., Pecher, I., Wallace, L., Klaeschen, D., 2016. Splay fault branching from the
Hikurangi subduction shear zone: Implications for slow slip and fluid flow. Geochemistry, Geophysics,
Geosystems 17, 5009-5023.
Plaza-Faverola, A., Vadakkepuliyambatta, S., Hong, W.L., Mienert, J., Bünz, S., Chand, S., Greinert, J., 2017.
Bottom-simulating reflector dynamics at Arctic thermogenic gas provinces: an example from Vestnesa Ridge,
offshore west-Svalbard. Journal of Geophysical Research: Solid Earth.
Portnov, A., Vadakkepuliyambatta, S., Mienert, J., Hubbard, A., 2016. Ice-sheet-driven methane storage and
release in the Arctic. Nature communications 7.
Riboulot, V., Thomas, Y., Berné, S., Jouet, G., Cattaneo, A., 2014. Control of Quaternary sea-level changes on gas
seeps. Geophysical Research Letters 41, 4970-4977.

Riedel, M., Wallmann, K., Berndt, C., Pape, T., Freudenthal, T., Bergenthal, M., Bünz, S., Bohrmann, G., 2018. In
situ temperature measurements at the Svalbard Continental Margin: Implications for gas hydrate dynamics.
Geochemistry, Geophysics, Geosystems 19, 1165-1177.
Roy, S., Senger, K., Braathen, A., Noormets, R., Hovland, M., Olaussen, S., 2014. Fluid migration pathways to
seafloor seepage in inner Isfjorden and Adventfjorden, Svalbard. Geological controls on fluid flow and seepage
in western Svalbard fjords, Norway. An integrated marine acoustic study.
Schiffer, C., Tegner, C., Schaeffer, A.J., Pease, V., Nielsen, S.B., 2018. High Arctic geopotential stress field and
implications for geodynamic evolution. Geological Society, London, Special Publications 460, 441-465.
Schneider, A., Panieri, G., Lepland, A., Consolaro, C., Crémière, A., Forwick, M., Johnson, J.E., Plaza-Faverola, A.,
Sauer, S., Knies, J., 2018. Methane seepage at Vestnesa Ridge (NW Svalbard) since the Last Glacial Maximum.
Quaternary Science Reviews 193, 98-117.
Sibson, R.H., 1994. Crustal stress, faulting and fluid flow. Geological Society, London, Special Publications 78, 69-
84.
Skarke, A., Ruppel, C., Kodis, M., Brothers, D., Lobecker, E., 2014. Widespread methane leakage from the sea
floor on the northern US Atlantic margin. Nature Geoscience 7, 657-661.
Smith, A.J., Mienert, J., Bünz, S., Greinert, J., 2014. Thermogenic methane injection via bubble transport into the
upper Arctic Ocean from the hydrate-charged Vestnesa Ridge, Svalbard. Geochemistry, Geophysics,
Geosystems.
Steffen, H., Kaufmann, G., Wu, P., 2006. Three-dimensional finite-element modeling of the glacial isostatic
adjustment in Fennoscandia. Earth and Planetary Science Letters 250, 358-375.
Svensen, H., Planke, S., Malthe-Sørenssen, A., Jamtveit, B., Myklebust, R., Eidem, T.R., Rey, S.S., 2004. Release of
methane from a volcanic basin as a mechanism for initial Eocene global warming. Nature 429, 542-545.
Turcotte, D., Ahern, J., Bird, J., 1977. The state of stress at continental margins. Tectonophysics 42, 1-28.
Turcotte, D.L., Schubert, G., 2002. Geodynamics. Cambridge University Press.
Vanneste, M., Guidard, S., Mienert, J., 2005. Arctic gas hydrate provinces along the western Svalbard
continental margin. Norwegian Petroleum Society Special Publications 12, 271-284.
Vogt, P.R., Crane, K., Sundvor, E., Max, M.D., Pfirman, S.L., 1994. Methane-generated (?) pockmarks on young,
thickly sedimented oceanic crust in the Arctic: Vestnesa ridge, Fram strait. Geology 22, 255-258.
Wallmann, K., Riedel, M., Hong, W., Patton, H., Hubbard, A., Pape, T., Hsu, C., Schmidt, C., Johnson, J., Torres,
M., 2018. Gas hydrate dissociation off Svalbard induced by isostatic rebound rather than global warming.
Nature communications 9, 83.
Westbrook, G.K., Thatcher, K.E., Rohling, E.J., Piotrowski, A.M., Palike, H., Osborne, A.H., Nisbet, E.G., Minshull,
T.A., Lanoiselle, M., James, R.H., Huhnerbach, V., Green, D., Fisher, R.E., Crocker, A.J., Chabert, A., Bolton, C.,
Beszczynska-Moller, A., Berndt, C., Aquilina, A., 2009. Escape of methane gas from the seabed along the West
Spitsbergen continental margin. Geophysical Research Letters 36, 5.
Zoback, M.D., Zoback, M.L., 2002. 34 State of stress in the Earth's lithosphere. International Geophysics 81, 559-
XII.

---

## Author Response (AR3)

Dear Editor,

We are thankful for this round of constructive and highly relevant reviews that helped us to improve the manuscript significantly. We hereby resubmit the manuscript for your consideration. Please find below the answers to each raised point by the referees and how the changes were implemented. The main changes in the manuscript can be summarized as follow:

- We have performed additional sensitivity tests for the geometry of the ridges in the model. The result is included in the supplementary, this include an additional figure. The main outcome of the additional tests is also presented in the main text.
- We have changed the paper structure to show a clear separation between results and discussion.
- The section where the model results is presented (5.1) includes now a paragraph that develops on the assumption of coupling between deep crustal and near surface deformation.
- We explain in the introduction the use of the terminology active vs. inactive pockmarks and refer to the presence or not of acoustic flares when possible.
- The discussion has been significantly extended: we developed further the analysis of modelled tectonic stress in the context of the total state of stress in the region; and we present a more substantial description of how we envision the evolution of present day and past seepage activity along the VR (including the mechanical relation between gas chimneys and faults). We close the discussion by placing our study in a context of global implications.
- All the specific comments have been addressed.
- Several figures were edited to support implemented changes. A new inset was included in former figure 3. A new figure is included in the result section to illustrate the origin of the tensile zone that extends towards the eastern Vestnesa Ridge. The figure numbers was updated accordingly. A figure for illustrating the model geometry was re-inserted in the appendix (we believe we lost that figure in the previous submission).
- Abstract, Introduction and conclusion sections were edited accordingly.

Sincerely,

Andreia Plaza-Faverola and Marie Keiding

Referee #1

In this contribution the authors are trying to bring together tectonic modeling (stress/strain) and observations on fluid seepage at Vestensa, off Svalbard. Their model suggests a region of tensile stresses that mostly coincides with a region of actively degassing pockmarks on the Vestnesa Ridge, juxtaposed to zones of strike-slip stress and inactive pockmarks.

I am reviewing this contribution after a first initial set of reviews by different reviewers was completed. I am judging my response on (a) the scientific material presented in the revised manuscript and (b) the earlier comments made by one reviewer on the applicability of the model used.

I find the approach presented in this manuscript intriguing, results rather interesting, but remain skeptical on the conclusions drawn. The paper needs some modifications, especially more discussions and at least one other set of tests if the zone of tensile stress is indeed robust in its location by modifying plate boundary geometries (which can be drawn differently than what authors show).

A few general remarks that I think would help in getting this paper published:

(a) Pockmarks on the southern drift. The authors show faults mapped south of the MTF, where seismic and bathymetric data show abundant chains of pockmarks. Those pockmarks are inactive in the terminology of the authors (i.e. inactive= no degassing). I suggest the authors augment their discussion on how to explain those southern pockmark chains as well. The orientation of principle stresses is strikingly different south and north of the MTF, relative to mapped fault-orientations (parallel, perpendicular) and earthquake focal mechanisms. Why is this not important as well?

Answer: thanks for pointing out this. The original focus of the paper was entirely on the Vestnesa Ridge seepage system (we wanted to test the hypothesis put forward in Plaza-Faverola et al., 2015). Also Vestnesa Ridge is much better studied than any other sedimentary ridge in the area. Having said that, we reckon that the correlation of these areas with the modeled stress fields makes sense.

 *Changes: We have included the location of these pockmarks in the discussion. Projection of pockmarks south of the MTF and right north of it where acoustic flares have not been documented as for the eastern Vestnesa ridge (i.e., Waghorn et al., 2018; Johnson et al., 2015) and discuss the spatial correlation of these features with the predicted stress field. We do emphasize that the features are less well investigated and a poorer control exist on the activity of this system. Ongoing studies by colleagues will provide more information about the structural setting in these regions.*

(b) Why only use degassing as sign of "activity"? Very recent heat-flow surveying (>100 measurements) taken across the pockmarks show that pockmarks may be still "active" even without gas discharge, as they can show a strong and localized heat-flow anomaly to the pockmark depressions themselves (and that occurs north and south of the MTF and the zone of tensile stress regime). The authors are aware of this study and data as well as preliminary results from the heat-flow study are available through the cruise report of expedition MSM57 and can be downloaded from the University of Bremen website.

Answer: We are aware of this recent heat flow study, yes. This is of course a relevant point in the sense that the terminology used is crucial. Despite anomalous heatflow measured nearby fault planes and pockmarks in the region (also Crane et al., in the late 90s show this), there is sufficient bubbling of gas only at discrete sites. The University of Tromsø goes to the region sometime 2 times a year, and vertical acoustical anomalies in the multibeam and echo-sounder profiles (so call acoustic gas flares) are observed systematically on the few pockmarks on the astern Vestnesa Ridge segment. The pockmarks at the Svyatogor ridge and north of the MTF have been surveyed also several years (despite having received much less attention so far). Here, acoustic flares have also not been observed. The terminology we use then uses "active" for sites where seepage is enough to cause a visible anomaly in water acoustic data. The inactive adjective does not rule out advection of fluids toward the near-surface nor degassing to a certain but minor degree compared to the "active" sites.

*Changes: we have clarified this through the text and figures. We also use the direct formulation of "documented acoustic flares" when possible.*

(c) Degassing could be completely unrelated to the deep-rooted stresses – microbial activity rates are changing; there is coupling between ocean phenomena and the sediment with free gas phases and gas hydrates (e.g. seen in tidal modulated gas fluxes or decadal oscillations). I think, using degassing alone is "too short" an argument to investigate the relationship between stress and the fluid flow regimes.

*Answer and changes: We have provide additional background on the fluid flow and seepage system along the ridge to make clear why current changes in sea-level and temperature would not be sufficient to explain solely the dynamics in this deep marine system. Yes, microbial methane generation is always a contribution to the seepage system. Along the Vestensa Ridge thermogenic gas (methane, ethaen, propane, butane, etc) have been sampled in head space samples from gravity cores as well as from water column and gas hydrate samples (e.g., Smith et al., 2014). The fluid flow systems is well documented to be associated with faults that extend close to the seafloor; gas chimneys are indeed following the fault planes. Petroleum modeling suggest that a thermogenic gas source may have been charging the system since 2 Ma ago (Knies et al., 2018). In this paper we rely on all these studies to assume that the advection of thermogenic/warmer fluids into the near-surface is a key player in gas hydrates and seepage dynamic in the region. Whether there is in addition seepage controlled by local microbial methane generation and to a small degree gas hydrate dissociation, is not so critical for this study. Regardless the source of the gas the behavior of faults can be modulating gas release.*

(d) Sediment drift bodies such as the Vestnesa Ridge are heavily dynamic and evolve quickly. Topography is steep across the ridge and vertical stresses thus change fast. While the authors do say and have indicated in the revised version of their manuscript that they do not reject other causes (and associated stresses) of the difference in vent activity, but I find the outcome or limitation of discussion only on the zone of tensile stress rather weak overall, given this very dynamic and regionally fast-changing zone of study.

*Answer and changes: we have now significantly extended the discussion to present clearer why the proposed tectonic explanation is so relevant for the study area and we provide summary of observations (large number of cross disciplinary studies) that points toward a regional tectonic mechanism.*

(e) The geometry of the plate boundaries, especially the MTF, seems to be fixed in all the discussion, but I could easily draw the MTF at a different angle (e.g. rotated by 5 degrees) by just looking at the bathymetry. Would this not affect the zone of tensile stress regime? The authors did look at parameter variations and assess effects in the actual zones of stresses, but they never changed the geometry. Maybe test quickly, if this is important or not?

*Answer and changes: Thanks for this suggestion. We carried out additional tests of the geometry by rotating the MTF as well as the MR and KR. The geometry does influence the shape or position of the tensile zone. However, the zone persists approximately in place. We have included results of diverse tests on geometry in the supplementary material. We also clarify, by including a new figure, how the tensile zone results due to the interaction between the spreading force from the Molloy and Knipovich ridges.*

(f) Coupling between crustal deformation and sediment load seems to have been assumed as 100%. Is that acceptable? Could those regimes be decoupled, i.e. the ridge-push is doing its own thing and sediment drape from contourites deposited during glacial/deglacial phases react to other forces?

*Answer and changes: We agree that this is a crucial and exciting question related to the topic. Is there a full coupling between crustal deformation and near-surface deformation of quaternary sediments? We discuss our results assuming a certain degree of coupling, as pointed out by the referee. We argue the near-surface faults reflect coupling (we interpret them as tectonic faults). We added a paragraph in section 5.1 with arguments supporting the assumption of a certain degree of coupling. We also touch upon the implications of assuming the coupling for the current debate about activity across passive margins.*

Overall, the manuscript has its contribution that is interesting and intriguing, fully worth of getting published at some point; but as of yet, it has too little result and discussion to be fully acceptable. After the initial revision, the authors modified and restated many points in their discussion and conclusion but need to go even further. I suggest a more diverse discussion taking some of my points above into account and possibly assess the "error" in the stress-regime mapping a step further.

Then, what are the larger implications of such process? Could you maybe name other regions where similar processes occur? What other studies would help assess this phenomenon further? Borehole stress measurements? Are there maybe wire-line image log (from resistivity) data from the old ODP boreholes to look at the borehole stress by mapping breakouts or fractures? Could you use the shear-wave splitting technique to look at the stress components by doing a polarization analysis of the first arrivals (e.g. Tonegawa et al., 2016)? Are those changing across Vestnesa and to the Svyatogor Ridge south of the MTF?

Answer: great suggestions here, thanks. We keep this in mind for further work on the topic. We have indeed on-going OBS experiments in the region. It would be interesting to further discuss the approach in the Tonegawa et al paper.

*Changes: we added a paragraph on the discussion to place the outcome of the study into a global context, emphasizing other known seepage systems where a stress control can be applicable.*

Tonegawa, T., Obana, K., Yamamoto, Y., Kodaira, S., Wang, K., Riedel, M., Kao, H., Spence, G.D., 2016. Fracture alignments in marine sediments off Vancouver Island from Ps splitting analysis, Bulletin of the Seismological Society of America, 10.1785/0120160090.

Referee #2

Dear Authors,

Understanding flow localisation patterns and location is a challenging entreprise. The submarine features leading to the expulsion of methane and other gases are widely reported, but currently the mechanisms remain still very unclear and discussed. In this context, this study tries to address a challenging point, constraining the flow localisation in Western Svalbard based on a analytical mechanical model of the complex lithospheric regional layout.

Although the model used seems to be correctly applied, the major simplifications and hypotheses the authors make in order to use it significantly limit the conclusion that can be drawn. This point was also discussed and criticised in the past round of revisions.

I will propose some broad suggestions that may enhance clarity and readability of the article, then address some specific amendments.

> Since the aim of this study is a modelling exercise, it would be nice to have a more rigorous model description. The model configuration is not very clear. It would be very helpful for the reader to see on the figure the initial and boundary conditions. On Fig. 4 as well, it would be very helpful to have these information present, as well as the highlight of the Vastness ridge.

Answer: We agree. In a previous version of the manuscript we had included a figure showing the model configuration, but that somehow slipped out.

*Changes: We added in the appendix a figure showing model configuration, and added spreading vectors to Figure 1. The crest of the Vestnesa Ridge is shown by the seismic line, as described in the caption.*

It would be very helpful for the reader to have a more structured result dissection and discussion. Maybe the authors could expose the full set of hypotheses and more quantitative facts in a first step, and then discuss and present their interpretation regarding the various aspects.

*Changes: we have changed the structure to have separated result and discussion sections. Please see above for details about the changes in the discussion. The discussion should reflect better what the state-of-knowledge is about seepage in the region and where does the importance of the tectonic hypothesis we present reside.*

Also, I would ask the authors to clearly distinguish between tectonic stress configuration optimal for enhanced fluid motion (extension regime) and actual mechanisms that may lead to flow focussing within chimneys and pockmarks. Fault planes will fail to produce pipe like features from a mechanical point of view.

*Changes: Thank you for rising this point. The relation between the chimneys and the faults/fractures is intriguing. Although the mechanical evolution of the gas chimneys escape the scope of the present study we extended the discussion about the spatial relation between faults and chimneys and touch upon the mechanical requirements to form a chimney. We offer a solution for a close spatial link between these two features.*

> The more specific comments are following:

L. 60-63: elastic stresses resulting from strain are instantaneous, thus I would be very careful using them to explain long term evolution and variations. They may be related to viscous creep rather than to purely elastic medium.

A: Thanks for pointing this out. In particular within the upper 50 m of sediments, the most recent interval through the gas chimneys towards the astern part of the ridge, have undulation features that remind features described in creep zones at other margins. However, there is a clear change with depth (from ca. 150 m) where the deformation is not expressed as chimneys but appear as near-vertical fault planes (documented in Plaza-Faverola et al). This paragraph was modified.

L. 73: please precise which fluids in the fluid dynamics you refer to.

A: done

L. 105-110: it is difficult to get the importance of the regional geology in the way it is brought to the reader. It would be very interesting to read about how these different geological formation impact the model, and the results.

A: The first section in the discussion links to the regional geological setting. The last 2 paragraphs were inserted in the discussion about the potential effect of tectonic stresses on near-surface faults.

L. 125-126: The last sentence is difficult to get, please rephrase.

A: done

L. 130: the GHSZ does not seem to be reported in Fig. 2, but rather in Fig. 3 ?

A: corrected

L. 134-138: What is the importance of those facts with regards to the proposed work ?

A: We have added a new reference here and elaborated on the idea for clarification. Thanks for pointing this out! The importance is that although the beginning of seepage is suspected to coincide with the onset of glaciations, the periodicity of the events does not correlate directly with either glacial or interglacials, but it seems to be more randomly controlled.

Section 4: Modelling approach. This section is about the aim of the paper, modelling. Thus, it would greatly benefit from enhanced details and a more rigorous model description, with regards to initial and boundary conditions especially.

A: A missing figure in the previous submission was re-inserted (appendix) and slightly edited. The appendix and the table in the supplementary material shall provide the necessary details about the modeling parameters.

L. 173: agreeing on the usage of a very simplified model, orientation of the stress field may be valid to interpret from the model, but I would be very careful regarding the magnitude of stresses, even if relative.

A: We reformulated for clarification. What we consider is rather the orientation of the principal stresses to determine the stress regime.

L. 210-213: Long and unclear sentence

A: Corrected.

L. 214: please precise style of principal stress

A: done

L. 218: please consider to rephrase avoiding the usage of "thing"

A: corrected. Apologies for that ☺

L. 219: please consider rephrasing as following "… provides a correct first order prediction of the stress field ORIENTATION in the upper crust…"

A: done: orientation and type

L. 221: consider the preferred usage of section number rather than below / above …

A: corrected

L. 236: consider rephrasing "thicker sediment thickness" to make it more clear

A: corrected

L. 240: what is seismic definition ?

A: changed to: delineation of faults in the seismic

L. 268: Gravitationnel stress would not only induce the suggested stress, but a complex pattern.

A: Right, the discussion was re-structured and this part was reformulated.

L. 292 what unit is mm/a ?

A: millimeters per year. For some reason the glacial isostatic modeling folk and also in chronology it is often used "a". Must be from the latin annos. We changed to year

L. 234-235: Please make it clear to distinguish between tectonic stress that may enhance fluid circulation in specific location from the actually mechanism that will result in fluid localisation within pipes or chimneys. Faults and fractures will not be able to produce tubular like features, based on mechanics.

A: Thank you for pointing out this. Thinking about the distinction between chimneys and faults in terms of mechanics of fluids lead to further discuss about the timing of faulting vs. chimney formation. We have elaborated this idea further and introduced a couple of additional references.

L. 335: consider replacing above by the appropriate section number.

A: changed accordingly through the entire manuscript.

Fig. 2: missing BSR in legend ?

A: corrected

Fig. 4: add boundary conditions, highlight the Vestnesa Ridge

A: done

Based on the aforementioned arguments, I would suggest to publish this article but asking minor revision.

[revised manuscript text omitted]

**5.2 Distribution of fault and seepage activity along the Vestensa Ridge with respect to modelled**

**tectonic stress**

High-resolution 3D seismic data collected on the eastern Vestnesa Ridge revealed sub-seabed NW-SE oriented, near-vertical faults with a gentle normal throw (< 10 m). In this part of the Vestnesa Ridge, gas chimneys and seafloor pockmarks are ca. 500 m in diameter. On structural maps extracted along surfaces within the gas hydrate stability zone (GHSZ) gas chimneys project over fault planes or at the intersection between fractures (Fig. 2).

A set of N-S to

NNE-SSE trending faults  outcrop at the seafloor at a narrow zone between the Vestnesa Ridge and the northern termination of the Knipovich Ridge (Fig 1, 2). These faults could be genetically associated with the faults on the crest of Vestnesa Ridge (Plaza-Faverola et al., 2015; Fig. 2), but they have also been suggested to indicate ongoing northward propagation of the Knipovich rift system (Crane et al., 2001;Vanneste et al., 2005).

[revised manuscript text omitted]

The intriguing, potential correlation between tectonic stress fields from mid-ocean ridge spreading and seepage
evolution along the Vestnesa Ridge, is an intriguing result that adds to the current debate about the quietness of
passive margins and the relation between deep crustal and near-surface deformation (Fejerskov and Lindholm,
2000;Fjeldskaar and Amantov, 2018;Lindholm et al., 2000;Olesen et al., 2013;Stein et al., 1989).

**67- CONCLUSIONS**

Analytical modelling of the stress field generated by oblique spreading at the Molloy and Knipovich ridges in the
Fram Strait, suggests that tectonic forcing may be an important factor controlling faulting and seepage
distribution along the Vestnesa Ridge, off the west-Svalbard margin. Other important sources of stress such as
bathymetry and lithospheric bending, contributing to the actual state of stress off Svalbard, are not considered in
the modelling exercise presented here; thus, we cannot quantitatively assess whether ridge push has a dominant
effect on seepage activity. However, provided a certain degree of coupling between our analysis of how crustal
and near-surface deformation, it is plausible that stresses from plate spreading may affect the behaviour of
mapped Quaternary faults along the Vestnesa Ridge. Spreading at the Molloy and Knipovich ridges leads to a
spatial variation in the tectonic stress regime exerted along the Vestnesa Ridge, that may favour fluid migration
through pre-existing faults and fractures on the. The eastern Vestnesa Ridge is under a stress regime that is
favourable favours for fluid migration through pre-existing faults and fractures. on the eastern Vestnesa Ridge

[revised manuscript text omitted]

---

## Author Response (AR4)

Dear Editor,

We are thankful for bringing the review of this study this far. All this work has improved the manuscript significantly and we highly appreciate it. We have addressed all the points by the referees. Please see below the list of addressed comments.

With kind regards,

Andreia Plaza-Faverola

Ref 1:

line 423 (page 14): please delete "thus" from the sentence. It seems out of place.

A: corrected.

line 445 (page 15) should 'maximums' not be 'maxima'?

A: isn't "Maximums" becoming the new standard plural of "maximum"  ☺. "Maxima" was commonly used in the past.

Ref 2:

I congratulate with the authors for the work done. This is a first important step to assess the relevance of tectonic stress propagation in the sedimentary cover of continental margins in controlling fluid migration in the long term. I acknowledge the work done to improve the manuscript according to the reviews received in the first round. I add below a few overall minor remarks and suggestions addressing some aspects not covered before.

Line 219. Earthquakes focal mechanisms. In the introduction (line 94), you state that earthquakes focal mechanisms provide poorly constrained stress vectors. However, in following analysis the focal mechanisms are used as a robust evidence of the stress field, so that they can be used to validate the results of the model. I wander why the rather clear evidence from focal mechanisms was not properly used. Perhaps the initial statement in the introduction should be smoothed?

A: Thank you for identifying this. We have changed the sentence in the introduction for coherency.

Line 232 and following text. One important aspect of your analysis, somehow not properly valued in the paper (although you address it in the discussion), is the role of compaction disequilibrium in the sedimentary column as a factor of weakening the strength of the sediments and therefore allowing the rupture to propagate upwards. I have a few arguments here that could be incorporated in the text with the purpose to give even more emphasis to the role of undercompaction:

You state rather vaguely here that the sedimentary column below the Vestnesa Ridge is not expected to be highly consolidated. I would not use the term 'highly consolidated'. Sediments are either normal- or over- or under- consolidated. Over consolidation results from the maximum stress experienced by the sediment being higher than the present stress. This occurs only in the case of subaqueous erosion (tectonic or sedimentary), or exhumation of a marine sedimentary sequence, which is not your case. In a sedimentary sequence experiencing continuous sedimentation, like the Vestnesa sediment drift, you either have normal consolidation (if excess pore water induced by sedimentation can dissipate without building overpressure) or underconsolidation (in the case you have a rapidly building total stress in consequence of rapid sedimentation, or increase in the volume of the pore fluids by, in your case, volumetric expansion of methane).

In this case your pore pressure may increase above hydrostatic, with decreasing effective stress (decreasing strength), to the point of reaching the total stress. Above this limit you may have hydrofracturing and sediment inflation. (I think this is the process you describe citing " the minimum effective stress is negative in Line 428. So hydrofracturing in the sedimentary column may be induced independently, without the crustal stress contribution, by a combination of increasing sedimentary load and increasing pore fluid volume. Of course in your case you are adding another factor that is the propagation of the crustal stress into the sedimentary column. In your case this contribution of tensile stress is towards a decrease of the total stress on the grains in favor of underconsolidation. I see you refer to this in Line 425. Later in the discussion you report a more important details, like the sedimentation rate. It would be important to know also the lithology (sampled or inferred). This is because it is generally assumed that in low permeability sediments (clay-rich formations), a sedimentation rate of 1 mm a-1 is enough to produce a situation of underconsolidation alone (Rubey & Hubbert 1959; Fertl 1976). You are not too far from this value below the crest of the ridge. I would rephrase your sentence in Lines 320-322 not I the sense of a stress build up, but rather a strength decrease due to compaction disequilibrium.

I am mentioning this because I think you should make even more clear in your discussion, the role of the consolidation state in the Vestnesa Ridge sedimentary sequence, saying that pore fluid overpressure (decreasing strength) may results from rapid sedimentation rate and fluid volume increase (gas expansion), below the crest of the ridge (the depocentre). The decreased effective stress in this area is a good explanation of the reason why you find the gas expulsion features and the faults here.

RUBEY, W.W. & HUBBERT, M.K. 1959. Role of fluid pressure in mechanics of overthrust faulting: II. Overthrust belt in geosynclinal area of western Wyoming in light of fluid-pressure hypothesis. Geological Society of America Bulletin, 70, 167–206

FERTL, W.H. 1976. Abnormal Formation Pressures. Implications to Exploration, Drilling, and Production of Oil and Gas Resources. Developments in Petroleum Science, 2. Elsevier, Amsterdam.

A: *Thanks again for raising this issue. This is a complex aspect of the system. The degree of consolidation of the sediment in the region remains uninvestigated. We have modified the sentence in line 232 to indicate this instead of speculating about the compaction.*

*In lines 320-330 we have included information about the lithology reported from gravity cores. We have also modified our statement to hint on the potential effect of under compaction (due to high sed rates and sediment grain size but also due to tectonic forcing) on effective stress. However, after revising the papers suggested by the referee and other papers studying consolidation of deep marine sediments (e.g., Buchan and Smith 1999), we realized that the upper sediments on Vestnesa may be under consolidated due to fast sedimentation and increase in pore fluid volume (as the referee points out) but on the other hand, the presence of authigenic carbonate and gas hydrates may favor over consolidation (or shift the consolidation curve on the opposite direction) and somehow compensate the abnormal effect on the compaction curve. This is a topic to be investigated in more depth, we hope that future experiments in the region will help in this matter. Nevertheless, the conditions for undercompaction and weakening of the sediment would be given along the entire ridge. Actually the highest sedimentation rate is towards the western segment. Hence, a decrease in the effective stress due to horizontal forcing in a tensile regime is still the best explanation for the focused seepage on the eastern segment.*

*We added emphasis with one sentence in the conclusion, to the importance of the effect of overpressure on decreasing the horizontal stress and favoring fracturing and dilation of existing faults particularly under the tensile stress regime at the eastern part of the ridge.*

Line 419. The role of gas hydrates in making the faults and fractures less permeable to fluid migration is very important. I suggest to cite and use the results of the paper:

Madrussani et al., 2009. Gas hydrates, free gas distribution and fault pattern on the west

Svalbard continental margin. GJI. doi: 10.1111/j.1365-246X.2009.04425.x

Where the authors provide geophysical evidence (Vp, Vs and attenuation) of the role of gas hydrates in decreasing the permeability of the faults and driving lateral gas migration below the base of the GHSZ.

*A: Great suggestion, thanks. We have included a reference to this work, we evoke their models as example of other areas where fluid dynamics at the base of the GHSZ due to sealing of faults by hydrates result plausible.*

Line 498. The meaning of bathymetry as a source of stress is, at least to me, a bit unclear. I understand you mean gravitational forces along a slope. Is it so? Is stress induced by bathymetry coded?

*A: We changed to gravitational forcing. We are not aware of any documentation of the forcing by bathymetry across the Svalbard margin. We hope this will be included in future models.*

Minors:

Line 243 replace "gentle" with "small"

A: corrected

Line 288. Replace "understood" with "considered"

A: corrected

Line 312. replace "nearby" with "focusing around"

A: corrected

[revised manuscript text omitted]